# A Multi- Mediation Analysis on the Impact of Social Media and Internet Addiction on University and High School Students’ Mental Health Through Social Capital and Mindfulness

**DOI:** 10.3390/ijerph22010057

**Published:** 2025-01-02

**Authors:** Girum Tareke Zewude, Tarikuwa Natnael, Getachew Tassew Woreta, Anmut Endalkachew Bezie

**Affiliations:** 1Department of Psychology, Wollo University, Dessie 1145, Ethiopia; getachew.tassew@wu.edu.et; 2Department of Environmental Health, College of Medicine and Health Sciences, Wollo University, Dessie 1145, Ethiopia; tarikuwa.natnael@wu.edu.et; 3Department of Occupational Health and Saftey, College of Medicine and Health Sciences, Wollo University, Dessie 1145, Ethiopia; anmut.endalkachew@wu.edu.et

**Keywords:** social media addiction, internet addiction, mindfulness, social capital, mental health, multi- mediation analysis

## Abstract

Introduction: Social media addiction (SMA) and internet addiction (IA) are increasingly prevalent, impacting mental health (MH) globally. This study investigates the mediating roles of mindfulness and social capital (SC) in the relationship between SMA, IA, and MH among Ethiopian high school and university students, contributing to the Sustainable Development Goal (SDG) 3 of good health and well-being. Methods: A cross-sectional study was conducted with 1160 university and 1473 high school students in Dessie, Ethiopia. Participants completed validated questionnaires assessing SMA, IA, mindfulness, SC, and MH. Structural Equation Modeling (SEM) with a multi- mediation Model (SMM) was used to examine the hypothesized relationships. Results: SEM revealed that both SMA and IA had a direct negative effect on mindfulness, SC, and MH in in both high school and university students. Notably, mindfulness and SC significantly and positively predicted MH, indicating their protective role against the negative effects of SMA and IA. Furthermore, both mindfulness or SC fully or partially mediated the relationship between SMA, IA, and MH, highlighting their crucial role in explaining the association. Conclusions: This study provides evidence for the crucial roles of mindfulness and SC in buffering the negative effects of SMA and IA on MH among Ethiopian students. The findings highlight the need for educational and health interventions that foster mindfulness and SC to enhance student mental health and promote a healthy digital environment. These results offer valuable insights for educators, health professionals, and policymakers in Ethiopia and other developing countries facing similar challenges.

## 1. Introduction

Globalization, driven by the widespread use of social media and internet access, has brought the world closer together, transforming it into a global village [1,2,3]. This digital system has made high school and university students’ learning and life easier, but the excessive use of the internet and social media may also cause mental, academic, and other health-related problems [4,5,6]. Globally, the number of Internet users climbed from 414 million in 2000 to 665 million in 2002, and by 31 December 2019, it had surpassed 4.574 billion [7]. Internet addiction is now becoming a major mental health issue among university and college students. For instance, in American 25.1% [8], in China it ranges from 10.6% to 13.6% [9], in Taiwan 15.3% [10], in Malaysian 36.9% [11], in Iranian 40.7% [12], in Japanese 38.2 [13], in Lebanon 16.8% [14], in Nepal 35.4% [15], in Jordan 40% [16], and in Turkey 1.6% [17], in Ethiopia it ranges from 35.2% to 85% [18]. Social media addiction is another possible worldwide problem that can have an impact on the mental wellbeing of the younger generation. This is because the majority of today’s youth squander precious time on social media due to their rising usage frequency. People become hooked on social media due to their constant need to use or log on to it, as well as the excessive time and energy they expend on it [19,20,21]. According to long-term cohort research conducted in the US, adolescents who spent time on social media for more than three hours a day had a twofold increased chance of suffering negative mental health outcomes, such as anxiety and depressive symptoms [22]. Young adults and their mental health have improved as a result of social media usage restrictions [23]. Another study has demonstrated that a four-week social media platform deactivation improves subjective well-being such as self-reported happiness, depression, and anxiety by approximately 25–40% compared to the effects of psychological interventions such as individual therapy, group training, and self-help therapy [24]. Based on research published [20], 12% of users of social networking sites suffer from social media addiction. Social media such as Telegram, Facebook, TikTok, Whats App, Instagram, Twitter, LinkedIn, and YouTube are the most popular media that have attractive and diverse spaces for online communication among students and the young generation [25,26]. Even if the use of the internet and social media is valuable for students, when they use them unhealthy or extensively, it becomes problematic to use the internet through these social media. Due to their working nature, university and high school students, who are the most productive parts of societies and represent future builders in any country, fall victim to social media and internet addiction, which affects their mental health. Mental health, which is our best resource and pillar for our thinking, feelings, and functioning in dealing with various life situations [27], plays a vital role in ensuring the dynamism and efficiency of any society. The mental health of the students is essential for raising their learning and scientific awareness [28]. Health promotion experts have become interested in the growing incidence of mental dysfunction and the associated costs to communities as a result of the shift in illness patterns in the modern world toward non-communicable diseases [27]. Addressing mental health issues in relation to internet addiction is therefore crucial, since addictive behavior on the internet has adverse effects such as depression and anxiety and puts people’s mental health at greater risk [29]. Most university students had addictive behavior toward the internet [30], and those who were addicted to the internet developed a higher risk of mental health than ordinary users of the internet [30], and excessive use led to psychological injury, mental health damage, and other health problems. This is because when they are used frequently, they can seriously harm a person’s lifestyle while neglecting their vital social interactions and general health.

Internet addiction (IA) and social media addiction (SMA) are two related but distinct issues that can have significant impacts on the mental health and well-being of young people [31]. While social media platforms and internet access offer opportunities for access to information, social connection, learning, and self-expression [31,32], they can also contribute to problems such as cyberbullying, social comparison, excessive screen time, declining academic performance, and deteriorating mental health [19,33]. Internet addiction (IA) refers to excessive and compulsive/problematic internet use as an addictive disorder that affects an individual’s overall health status and interferes with daily activities [34,35,36], whereas social media addiction (SMA) focuses on problematic social media use or social networking addiction, involves an excessive and compulsive use of social networking platforms such as Facebook, Instagram, Twitter, and Snapchat [37]. Research conducted in Saudi Arabia, China, South Korea, and Ethiopia has revealed high rates of internet addiction among high school and university students [38,39,40]. For example, studies in China and South Korea conducted on internet addiction found that adolescents reported a 13% and 22% rate of internet addiction [39,40], respectively. Research by [41] reported that university students reported higher mean scores in internet addiction than high school students. A study also conducted in Ethiopia found that internet addiction negatively and significantly predicted mental health and positive psychological capital [6]. These studies have shown that SMA and IA have negative impacts on mental health, positive psychological variables, academic performance, mindfulness, moral potency, sleep patterns, and overall well-being [19,42,43,44,45]. The detrimental effects of IA include depression, mental health disorders, a decreased sense of control, and a hindrance to sleep patterns [46,47]. Similarly, SMA has been associated with social anxiety, decline of academic performance, social capital and mental well-being, low self-esteem, mood and anxiety disorders, and cyberbullying [48,49,50,51]. Social media addiction affects student academic performance in the following ways: spending more time online and less time on study, interrupting students’ time management, and interfering with students’ work by distracting them and making them unable to stay focused, which may make it difficult for them to encode and remember what they are learning [52,53]. Prolonged usage of social media causes psychological and medical issues, including obesity [54] and mental comorbidities, disrupted sleep patterns and result in poor sleep quality, social isolation, loneliness, depression, sadness, suicidal thoughts, cyberbullying, online harassment, sedentary behaviors, and disorders related to alcohol and tobacco use, which further exacerbate mental health issues [55,56]. Excessive use of the IA and SMA affects their brain by triggering dopamine release, which can lead to dependency and withdrawal symptoms. Besides, SMA is positively associated with social anxiety [48], burnout and depersonalization [57], academic performance, and mental well-being [51]. Due to the nature of constant connectivity and real-time updates, social media creates a fear of missing out on experiences and compulsive checking behaviors. Mindfulness enhances self-regulation by increasing awareness of one’s thoughts, emotions, and behaviors, helping students recognize signs of addiction and manage their impulses to engage excessively with social media or the internet. This can reduce stress, anxiety, depression, and other psychological hazards. Social capital provides enhanced social support through networks of relationships and norms of reciprocity and trustworthiness that buffer against stress, loneliness, and other negative emotions linked to SMA and IA. These constructs are vital for developing interventions and preventive strategies aimed at mitigating IA and SMA and enhancing an individual’s overall well-being and performance [58,59]. These findings highlight the need to understand the complex relationships between SMA, IA, and social capital, mindfulness, and their impact on mental health and academic performance. To address the research gap, this study aimed to examine the connections between IA, SMA, social capital, mindfulness, and mental health. This study adopts a multi-mediation analysis.

## 2. Literature Findings of Social Media Addiction, Internet Addiction and Its Several Associations

### 2.1. The Relation Between Social Media Addiction and Mental Health

Social media has emerged as the leading online service, drawing the attention of millions of users worldwide. Due to the possibility of fast communication and access to a large amount of information disseminated swiftly [60,61]. However, scientific evidence shows that SMA affects people’s mental health [52]. For example, addicted social media users were associated with greater levels of depression and suicidal thoughts [62]. SMA is a psychological state of maladaptive dependence and compulsive use [63] and is overly concerned and strongly motivated to devote vast amounts of time and energy to using social media. The implications of SMA are significant [63] and overly concerned and strongly motivated to devote vast amounts of time and energy to using social media. The implications of social media addiction are significant. Research suggests low work performance [64], low life satisfaction [65], and high stress [66] to be some of the adverse outcomes. The majority of young people in today’s generation waste their valuable time on this platform due to increased usage frequency. The insatiable need to use or log on to social media, as well as spending so much time and energy on it, cause people to become addicted to social media [20]. According to long-term cohort research conducted in the US, adolescents who spent time on social media had an increased chance of suffering negative mental health outcomes [22] and social media usage restrictions have improved mental their health [23]. Another study has demonstrated that a social media platform deactivation improves subjective well-being such as self-reported happiness, depression, and anxiety [24]. Mental illness is a growing public health issue, has a greater impact on people’s health globally, and negatively impacts individuals’ quality of life at home, work, school, and in social settings [67]. Knowing the precise elements that enhance or deteriorate mental health is helpful for enhancing the efficiency of health promotion programs, predicting mental health declines, and comprehending the beginning and progression of illness and its symptoms, which can manifest as mood, cognitive decline, physical and emotional reactions, and interpersonal and psychological issues. Another study reported that users of social networking sites suffer from social media addiction [20].

Social media addiction is a current public mental health concern that involves uncontrollable use of the Internet for a prolonged period per day. It is also called pathological internet use, leading to psychosocial distress and significant impairment in health or interpersonal relationships [68].

In today’s world, college and university students highly use internet access for social media due to the better accessibility of smart phones and the internet, which helps the learning and teaching process. This internet use through social media helps students overcome their needs for exit exams, graduate admission tests, e-learning, research and project work, and their day-to-day education. Moreover, students use their internet for the purposes of social networks, entertainment, watching movies and news, checking email, using internet games, online shopping, maintaining relationships, and making new friends [69]. However, the use of social media and the internet excessively becomes problematic since it consumes their time and academic thought, resulting in academic failure, poor direct social interaction, decreased concentration ability, a negative affective state, and poor engagement in physical activity [70]. The extensive use of social media results in physical and social psychological problems such as psychiatric comorbidities, insomnia, depression, suicide ideation, anxiety, tobacco use disorders, alcohol use disorders, sleep disorders, and obesity [71,72,73]. Overuse of social media also fosters a climate that is advantageous for students, emphasizing inappropriate sexual exposure online, victimization from cyberbullying, excessive use of computers for academic purposes, and excessive use of computers for gaming purposes [74]. Social media addiction was related to sociodemographic factors like gender, smartphone use, and living area [13,75]. Men use Facebook and Twitter at rates of 16.4% and 26%, respectively, while women use them at rates of 8.3% and 13%, respectively [53]. Another study reported that girls are more likely than boys to visit Facebook and Instagram frequently [55]. Moreover, some common characteristics of university students, such as being unmarried, younger, having access to a smartphone, and being unemployed, make them problematic users of the internet on social media [7,31]. It is significantly increased with grade [76,77]. Male and senior students experience more problematic internet use than females and juniors [77]. In contrast, a study in Italy reported that adolescent females were more problematic internet users than their male counterparts [78]. The study also shows a negative association between parental norms about internet use, parent-adolescent contact, and teenage problematic internet use, while parental internet use was positively connected with teenage problematic internet use [79]. In sum, social media addiction is a valid predictor of mental health among both university and high school students. Thus, we proposed the following research hypothesis (RH):

**RH1.** 
*social media addiction negatively affects mental health among both university and high school students.*


### 2.2. The Relation Between Social Media Addiction, Mindfulness, and Social Capital

Students use their smart phones for social media and internet purposes due to their portability, speed, convenience, personalization, and aggregate use. Researchers [79,80] found that mindfulness and social capital are the most potential psychological resources that may safeguard mental health, reduce loss of relationships and loss of control, and have a protective role in social media addiction and internet addiction in adolescents [81,82]. These psychological resources are also the most important positive psychological resources that substantially support an individual’s healthy digital system and functioning while using both social media and internet access [83]. Mindfulness can be described as an intentional, present-focused awareness without judgement, and it involves directing one’s attention to the current experience in a receptive and open manner rather than dwelling on the past or worrying about the future [84,85,86,87], which helps to overcome social media addiction and safeguard the mental health of adolescents [86,87]. Unlike constructs like positive psychological capital, emotional intelligence, or even social cognitive theory, mindfulness does not rely primarily on cognitive processing or intellectual engagement; rather, it is characterized by a receptive, non-judgmental awareness and acceptance of the current experience as it unfolds [85]. Mindfulness is a self-regulation technique that places a strong emphasis on being aware of and paying attention to the present moment in order to improve emotional control and produce positive emotional outcomes [85,88,89,90,91]. This practice has been negatively related to social media addiction, stress, depression, and mental health [91,92,93], whereas it has been positively related to mental health, social capital, and positive outcomes on hedonic and eudaimonic well-being constructs [94].

Specifically, a study by Rostila et al. [83] found that smartphone addiction, such as cognitive preoccupation with one’s smartphone and compulsive patterns of use, was significantly associated with lower-level mindfulness. Similarly, ref. [92] revealed that more problematic social media use is associated with a lower level of mindfulness. A study by [95] found that social media addiction reduces mindfulness. In addition, various studies have reported the crucial role of mindfulness, such as promoting intercultural, interreligious, and inter-contemplative competencies, helping to address swiftly pathogenic factors, reducing social media addiction or problematic social media use [92,96,97], increasing positive emotions, reducing stress, and improving quality of life [98,99,100]. Remarkably, mindfulness has a protective role in the correlation between mental health and SMA in adolescents [82]. 

Social capital is another substantial positive social resource for high school and university students to overcome SMA and help foster a healthy digital system. A study by [101] found that social capital is defined as a personal relationship and the benefits that come with it, with some individuals interacting, forming bonds, and bridging. A study show that social media addiction is negatively associated with social capital bridging [79]. However, other studies confirmed that social capital played a key role in enhancing the COVID-19 response or adverse effects, including addiction [100]. Similarly, social capital and mindfulness have been shown to positively correlate, whereas perceived stress and social capital have been found to negatively correlate [101]. Such inconclusive arguments need further research in different cultural settings using advanced methods and a diverse population. The above scientific evidence leads us to consider that social capital and mindfulness as positive psychological and social resources play an important protective role in enhancing social media addiction and boosting mental health through a healthy digital system for high school and university students. Therefore, the substantial relationship between social media addiction social capital, mindfulness, and mental health needs more empirical investigation in comprehensive. Thus, we proposed the following testable research hypothesis:

**RH2.** 
*Social media addiction negatively affects social capital, mindfulness, and mental health among both university and high school students.*


### 2.3. Internet Addiction and Mental Health

Adolescents’ health and well-being are negatively impacted by excessive internet use [81], which has been linked to stress, sleep issues, and personality disorders [102]. According to a study conducted on Iranian students, social media use is linked to mental health problems like anxiety and depression [103]. A previous study also reported that increased internet use among college students is directly tied to a decline in the level of mental wellbeing and adjustment [104]. Internet addiction may lead to low psychological well-being due to the limited dimension of friendship [105]; isolation, stress, and loneliness are more common among students who are addicted [106]. Moreover, internet addiction is negatively and significantly related to general positive affect, emotional ties, and life satisfaction among adolescents [105]. It may cause students to spend less time talking with their families, experience more daily stress, feel lonelier and depressed [105], and experience social impairment that affects general well-being [107]. Additionally, internet addiction has been linked to psychological and behavioral health issues like poor physical health like sleep deprivation, exhaustion, unhealthy eating habits, and a lack of physical activity [108], anxiety and depression [109], and attention deficit hyperactivity disorder [110].

Studies have also demonstrated a correlation between internet addiction and social issues, including inadequate communication [111], heightened parental conflict [112], and addictions to online gaming, gambling, and pornography [113]. To enhance individuals’ well-being, mindfulness would be helpful to individuals’ conducive coping mechanisms while suppressing maladaptive coping strategies [114]. This is because internet addiction is mediated by mindfulness, and high mindfulness can alleviate the negative effects of internet addiction or a low level of psychological resources [115,116]. This means individuals with low mindfulness exhibit addictive smart phone use, and as the level of their mindfulness increases, their problematic phone use decreases [117,118]. 

Conversely, people who practice high mindfulness tend to use the internet less than people who practice low mindfulness [119]. This is because people who have high mindfulness levels frequently have optimistic and upbeat outlooks on life and the future, and they are better able to handle issues like problematic social media and internet use [119]. Therefore, we proposed the following testable research hypothesis:

**RH3.** 
*Internet addiction negatively affects mental health of the student population.*


### 2.4. Internet Addiction, Social Capital, and Mindfulness

A study found that extensive internet use among participants was associated with a decline in social life, an increase in feelings of hopelessness and loneliness, and reduced communication with family members at home. Specifically, as individuals spent more time online and accumulated less social capital, they had less time for direct interactions with friends, family, and colleagues [120].

For instance, internet use at work was strongly linked to decreased time spent with coworkers, while excessive internet use at home significantly detracted from the quality of time shared with family and friends. These findings suggest that greater engagement in online activities may lead to diminished opportunities for meaningful face-to-face connections, ultimately impacting individuals’ overall well-being.

Previous studies have demonstrated that an increase in internet addiction is associated with greater social distancing. Specifically, there is a negative correlation between social capital and internet addiction, indicating that higher levels of social capital can help prevent the development of online addiction [92,121]. This suggests that fostering strong social connections may mitigate the risk of internet addiction and its associated impacts on individuals’ lives. University students who are addicted to smartphones experience greater negative impacts on their social relationships, leading to weakened attachments to their family, close friends, and relatives [122]. Due in large part to their demanding academic programs and maladaptive health habits, university students are widely recognized as a vulnerable population susceptible to psychological health difficulties [123]. Nevertheless, Internet addiction (IA) remains an area of research that has received comparatively less attention. In today’s society, adolescents are particularly vulnerable to the negative effects of IA, which can adversely impact their moral character and psychological well-being [124].

Research has identified mindfulness and psychological capital as the most effective positive psychological tools for preserving ego strength and reducing levels of internet addiction [125,126,127,128]. These constructs are vital for developing interventions and preventive strategies aimed at mitigating internet addiction. Additionally, they play a crucial role in enhancing an individual’s overall well-being and performance, especially in cases of severe internet addiction [126,128].

Mindfulness is the discipline of being completely present and engaged in the present moment without passing judgment [127], which has been widely researched for its several benefits. These include buffering against internet addiction [128], improving mental health [6], reducing depression [123], and safeguarding ego strength [19]. A study found that mindfulness interventions have potential benefits in reducing internet addiction [19,128]. A comprehensive study by [126,128,129,130] found that internet addiction negatively and mindfulness positively predicted students’ mental health, and a negative association was found between mindfulness and internet addiction. Thus, both university and high school students who develop internet addiction may be less likely to have social resources and mindfulness skills. This led to the following testable research hypothesis:

**RH4.** 
*Internet addiction negatively affects social capital, mindfulness, and mental health among both university and high school students.*


### 2.5. Social Capital, Mindfulness, and Mental Health

Social capital is one of the main potential safeguards against internet and social media addiction. People’s mental health significantly improved when they had more social capital, such as cognitive social capital [130,131]. Social capital may have an impact on health outcomes by eliminating possibilities for unhealthy behavior and quickly promoting and disseminating health information [131].

Increased financial, medical, and transportation resources are made available to people with high social capital, which has been shown to improve general health [132]. Larger social networks may make it easier for a person to get social assistance, which improves health [118]. Furthermore, individuals with access to network members who engage in healthy lifestyles may seek information from these connections, which can reinforce and promote positive health behaviors [133]. Social capital, such as a strong social network and higher trust in neighbors, has a protective factor against depression [133].

People who report feeling socially isolated within their networks are more likely to experience depression than those who maintain a greater number of connections [119,120]. Social capital is positively correlated with psychological well-being, suggesting that a robust network of social relationships plays a crucial role in enhancing mental health outcomes. The second influential positive psychological resource is mindfulness, which promotes mental health and acts as a moderator between internet and SMA and mental health. The literature that is now available illustrates the connection between mindfulness and internet addiction, which has a significant impact on people’s social lives. The detrimental impacts of social media can be successfully mitigated by mindfulness [134]. It provides benefits to individuals by maintaining mental health [135]. According to studies by [134,135], it helps people by preserving mental health, raising subjective well-being, improving mental clarity and intensity of attention, boosting productivity, lowering stress and depression. Certain research has linked people’s mental health and mindfulness. For instance, studies show that high levels of self-esteem and low levels of social anxiety are significantly predicted by mindfulness [133,134,135,136]. Social capital also reduces anxiety and improves overall mental well-being [137,138]. Thus, we hypothesized that:

**RH5.** 
*Social capital and mindfulness positively affect mental health among both university and high school students.*


### 2.6. Theoretical Foundations and Research Hypothesis

The positive psychology theory proposed by Seligman [139], the social cognitive theory developed by [140], transactional model of stress and coping (TMSC) by [141]; and the cognitive-behavioral model of pathological internet use [142,143] serve as the theoretical foundations for this novel study. The theory of positive psychology, as developed by [139], has been associated with various phenomena such as internet addiction, social media addiction, mindfulness, social capital, and mental health. Consequently, positive psychology theory offers a practical framework for examining the role of mindfulness as mediator in the relationship among IA, SMA, and the mental health of students. Nowadays, due to its highly beneficial qualities, mindfulness construct had integrated with the positive psychology model [144], and mindfulness based programs and practices have been highly associated with many positive mental health and psychological outcomes [145].

Additionally, Albert Bandura’s Social Cognitive Theory [140] explains that regulating social activities, including healthy social media use and internet access, plays a crucial role in regulating social and moral behavior as well as essential resources for social functioning, fostering the development of social competencies and internal standards. In addition, the transactional model of stress and coping (TMSC) by Davis [141] posits that individuals’ response to stress involves transactions between the person and their environment and it could be relevant for understanding how mindfulness and social capital act as coping mechanisms in dealing with stressors related to IA and SMA [141]. To strengthen these ideas, a study by Rizzo and Alparone [36] found that adolescents experiencing greater feelings of loneliness during isolation had poorer mental health and a lower quality of life. This underscores the importance of social capital. The theory also provides the best framework for investigating how individuals’ coping strategies, including mindfulness practices and social capital (social support networks), influence their experiences with addictive behaviors [141].

Finally, a cognitive-behavioral model of pathological internet use (PIU) by [142] theorizes that internet addiction links cognitive symptoms such as ruminative cognitive styles, low self-worth, and social anxiety to the development of internet addiction, particularly in the context of social media use [142]. This model suggests that these cognitive factors can contribute to excessive and problematic online behaviors, affecting social capital by potentially reducing face-to-face interactions and impacting mental health through heightened feelings of loneliness, isolation, and diminished self-esteem, emphasizing the need for interventions that address these cognitive patterns to promote healthier internet use and improve overall well-being [6,142,143]. The theoretical models used in this study such as the Seligman’s positive psychology theory [139], Bandura’s social cognitive theory [140], the transactional model of stress and coping (TMSC) [141], and a cognitive-behavioral model of Pathological Internet Use (PIU [142] emphasizes that social networking, social capital and mindfulness are crucial components for enhancing mental health and reducing internet and social media addiction among university and high school students) [140,141,142,143]. These theories emphasize the importance of person-environment interaction, social net-working, being aware of the environment, social capital and mindfulness as positive psychological and social resources for reducing internet and social media addiction and promoting mental health.

By conducting an in-depth study into the mediating effects of social capital and mindfulness, this innovative study aims to offer valuable insights into the complex relationships among social media addiction (SMA), internet addiction (IA), mindfulness, social capital, and mental health amongst both university and high school students. The findings of this research will not only contribute to the existing literature but also have significant implications for policymakers, educators, and mental health professionals, shedding light on the potential positive impact of social capital and mindfulness in nurturing students’ digital mental well-being. Through its innovative approach, this study seeks to make a significant contribution to the field by addressing the gap (methodological, conceptual, and contextual) in knowledge surrounding the influence of social capital and mindfulness on students’ digital mental health. By uncovering the underlying mechanisms and pathways through which these factors operate, this research has the potential to shape effective interventions and strategies aimed at promoting healthy digital habits and overall mental well-being among students. By emphasizing the significance of mindfulness and social capital in the 21st century as important psychological skills within the context of digital environments, this study stands out as a novel and original endeavor. It recognizes the changing landscape of students’ lives, highly influenced by the digital realm, and seeks to empower individuals and institutions with evidence-based insights to navigate these complex associations. As a result, this study not only has academic relevance but also carries practical implications that can inform the development of policies and educational initiatives tailored to support students’ mental health in the digital era.

Limited research has been conducted in Ethiopia to investigate the relationship between internet addiction and mental health [6,143]. This study is novel in that, to the best of our knowledge, no previous studies have explored the role of multi- mediation analysis of social capital and mindfulness in the association among social media addiction, internet addiction, and mental health from a developing country perspective or cultural setting. Based on the theoretical frameworks outlined above, it is essential to investigate the potential impact of social media and internet addiction on high school and university students’ mental health, considering social capital and mindfulness as potential positive resources that may be depleted. To enhance the robustness of the study, it is recommended to utilize larger and more diverse samples. Therefore, we propose the following research hypothesis:

**Hypothesis** **6a.**
*social capital and mindfulness partially mediate the relationship between social media addiction and mental health, and internet addiction and mental health constructs.*


**Hypothesis** **6b.**
*social capital and mindfulness fully mediate the relationship between social media addiction, internet addiction, and mental health.*


In summary, based on the positive psychology theory [139], the social cognitive theory [140], the transactional model of stress and coping [141] and the cognitive-behavioral model of pathological internet use [142,143], the purpose of the present study was to test the mediation effects of social capital and mindfulness on the relationship between social media addiction, internet addiction and mental health of university and high school students. The proposed interrelationships are visually represented in the hypothesis model depicted in Figure 1:

## 3. Materials and Methods

### 3.1. Research Design

A large cross-sectional design with an associational approach is well suited to the current study to achieve the stated objectives. To achieve the stated objectives, we targeted both high school students and college students belong to age groups, with levels of maturity and responsibilities. High school students are usually younger. May be in a phase of exploration and social growth were social media’s crucial for building and sustaining peer connections. On the contrary college students are generally more concentrated on achievements. Preparing for their careers which could affect why they use social media and internet. The Ministry of Education’s move towards platforms and digital resources has had an impact on both high school and university students alike due to the shift in educational methods [6,19,144]. Understanding how students interact with the internet and social media has become essential for educators and policymakers as reliance on tools for learning continues to grow. Studying the variations in media and internet usage patterns and susceptibility to addiction among school and college students offers valuable insights into how these platforms impact students at diverse academic stages. The knowledge of these distinctions can aid in creating tailored interventions and support mechanisms to encourage habits among students.

### 3.2. Study Setting

The study’s target populations are Amhara Regional State public university undergraduate and five selected high school students in Ethiopia.

### 3.3. Sample and Sampling

The samples were selected from one public university and five selected high schools in the Amhara Regional State of Ethiopia. The university and high schools were purposefully selected due to their convenience for data collection and their large populations. The samples were then chosen using proportionate stratified random sampling. For instance, the high school student samples were selected across grades and genders (refer to Table 1) based on the population numbers. After clearly defining the strata, we utilized simple random sampling to select the samples. The sample comprised 2633 respondents, with a mean age of 19.3 years (SD = 2.03), ranging from 16 to 29 years of age. The sample encompassed 1160: 689 (59.4%) male and 474 (40.9%) female undergraduate graduate students at Wollo University in Dessie, Ethiopia and 873 (59.3) males and 600 (40.7%) females with a total of 1473 high school students from Dessie Town, Ethiopia (See details in Table 1). In this study, the researcher intends to use advanced structural equation modelling and psychometric measures (confirmatory factorial analysis, discriminant, and convergent validity) based on proposed general guidelines for absolute sample size: (1) small (*n* = 100); (2) medium, *n* = approximately 150; and (3) large (*n* > 200) [146]. Thus, it is often recommended that researchers have 200 cases or more for their study sample to achieve statistically stable estimates and fewer sampling errors [146]. As a result, in both planned studies, we adhere to this guideline.

### 3.4. Instruments

The reliabilities (Cronbach’s alpha, composite reliability) and validities (convergent, discriminant validity, construct validity using confirmatory Factor Analysis-CFA) of the scales were calculated and reported in this study.

#### 3.4.1. Demographics Questionnaire and Data Quality Controlling

The demographic questionnaire obtained information regarding sex, age, grade level, and batch (year) of the students. Additionally, to increase the accuracy of the data, there were four attention and honesty check items included based on scientific recommendations [147,148]. The first item, “Please choose ‘agree’ for this question”, was used to test whether the subject was careless or inattentive. If the subject did not choose the fixed response of “agree”, their data was excluded [147,148,149]. The next three items which were rated on a 4-point Likert scale from 1 (strongly disagree) to 4 (strongly agree)—“I answered all the questions truthfully”, “I never lied”, and “I never hid myself”—were intended to assess the honesty and truthfulness of the subject’s responses. If a person scored low (≤2 on the 4-point Likert scale) on any of these three items, their data was also excluded, as this would suggest a tendency toward self-reported deception [147,148,149]. By including these validity checks, the researchers aimed to ensure the truthfulness and reliability of the self-reported data collected through the demographic questionnaire.

#### 3.4.2. Bergen Social Media Addiction Scale (BSMAS)

The BSMAS used in this study (see Appendix A for both university and high school students) was a six-item developed by [150] based on Bergen Facebook Addiction Scale which includes six main elements (mood, salience, tolerance, modification, conflict, relapse, and withdrawal) of addiction proposed by [151]. The BSMAS is a single factor scored between very rarely (1) and very often (5) points. The scale was adapted to the Turkish culture by [152] and had a strong reliability and construct validity. The Cronbach’s α value of the scale was found to be 0.88. In the present study, the scale in both university and high school students demonstrated (α = 0.963, 0.962; CR = 0.963, 0.963; AVE = 0.810, 0.807), high internal consistency, construct reliability and discriminant validity respectively. The confirmatory factor analysis (CFA) fit values showed an acceptable fit in the current study for both university and high school students (χ^2^/df = 6.70, 6.71 TLI = 0.911, 0.904, CFI = 0.912, 0.920, RMSEA = 0.085, 0.88 (95% CI = 0.076, 0.043 to 0.098, 0.0.72), SRMR = 0.029, 0.33), respectively.

#### 3.4.3. Internet Addiction Scale (IAS)

An original instrument of the IAS developed by [153] later adapted by [6,143] to assess an individual’s excessive and compulsive internet use that interferes with daily functioning using a 17-item scale used in this study (see Appendix A for both university and high school students). The IAS was a seven-item response option ranging from Very Strongly disagree (1) to Very Strongly agree (7). The IAS is a total of 17 items measured with four dimensions and in the present study the internal consistency, construct reliability, convergent and discriminant validity respectively were: (a) Internet craving (α = 0.936, 0.934; CR = 0.936, 0.934; AVE = 0.746, 0.739), (b) Internet compulsive disorder (α = 0.913, 0.910; CR = 0.914, 0.911; AVE = 0.726, 0.717), Addictive behavior (α = 0.903, 0.900; CR = 0.905, 0.904; AVE = 0.704, 0.696), and Internet obsession (α = 0.931, 0.924; CR = 0.932, 0.926; AVE = 0.772, 0.756) and the total scale was α = 0.94. A construct validity of the measure using CFA found that the four-factor model looked to provide an excellent fit to the data (χ^2^)/df = 4.27, 4.53, TLI = 0.973, 0.968, CFI = 0.978, 0.973, RMSEA = 0.053, 0.061 (95% CI = 0.048, 0.051 to 0.058, 0.078), SRMR = 0.029, 0.030), respectively. In the previous study, the construct validity (acceptable) and reliability both Cronbach alpha and CR (ranged from 0.883, 0.883 to 0.927, 0.927) of the scale was also confirmed, respectively [6]. The Internet Addiction Scale negatively correlated with mental health [6] and mindfulness [19,154].

#### 3.4.4. Personal Social Capital Scale-8 (PSCS-8)

The PSCS-8 consists of 8 items (see Appendix A for both university and high school students) and each item is scored on a 5-point Likert ranging from 1 (a few) to 5 (a lot) originally developed by [155], the short form revised by Wang et al. [101] was used in this study. PSCS-8 is grouped into two dimensions: bonding and bridging. The reliability coefficients for the PSCS-8 scale in the previous study for two factors were: bonding (0.70), and bridging (0.73) and total (0.78) [101]. A sample item is “There is a special person who is around when I am in need”. The two factor PSCS-8 scale in the present study for both university and high school students demonstrated: bonding (α = 0.936, 0.942; CR = 0.936, 0.943; AVE = 0.786, 0.804), and bridging (α = 0.930, 0.906; CR = 0.930, 0.908; AVE = 0.768, 0.711), high internal consistency, construct reliability and discriminant validity respectively. Moreover, a confirmatory factor analysis (CFA) for both university and high school students (χ^2^/df = 4.78, 4.43, TLI = 0.949, 0.946, CFI = 0.965, 0.963, RMSEA = 0.067, (95% CI = 0.061, 0.051 to 0.098, 0.091), SRMR = 0.026, 0.032) was also conducted on the scale to check the construct validity and confirmed an acceptable range based on the conventional criteria’s of [156].

#### 3.4.5. The Five Facet Mindfulness Questionnaire Short Form (FFMQ-SF)

The FFMQ-SF developed by [126] and later modified by [127] was used in this study (see Appendix A for both university and high school students) aimed to assess mindfulness levels. The FFMQ-SF has five sub-scales and respondents’ responses are given on a 5-point Likert scale ranging from 1 (very rarely true) to 5 (almost always true). The original version of each sub-scales with four items: acting with awareness, describing, observing, non-judging, and non-reactivity internal consistency ranged from α = 0.69 to 0.85, which demonstrated acceptable reliability. In the present study, the five sub scales for both university and high school students: (a) acting with awareness (α = 0.952, 0.946; CR = 0.952, 947; AVE = 0.833, 0.816), describing (α = 0.946, 0.938; CR = 0.947, 0.940; AVE = 0.818, 0.797), observing (α = 0.947, 0.945; CR = 0.948, 0.945; AVE = 0.819, 0.812), non-judging (α = 0.908, 0.894; CR = 0.910, 0.896; AVE = 0.717, 0.685) and non-reactivity (α = 0.922, 0.905; CR = 0.927, 0.911; AVE = 0.761, 0.728) demonstrating high internal consistency, construct reliability and discriminant validity respectively. The construct validity of the FFMQ-SF using CFA demonstrated that the five-factor model for both university and high school students looked to provide an excellent fit to the current data (χ^2^/df = 4.38, 4.47 TLI = 0.973, 0.965, CFI = 0.977, 0.971, RMSEA = 0.054, 0.059 (95% CI = 0.050, 0.056, 0.058, 0.063), SRMR = 0.04, 0.035), respectively.

#### 3.4.6. Mental Health Continuum-Short Form [MHC-SF]

Mental Health Continuum-Short Form [MHC-SF] is the most widely used tool designed to evaluate the mental health status of an adolescent and youth used in our study (see Appendix A for both university and high school students). The main MHC-SF was used to evaluate the mental health of the participants [152]. The frequency of enjoyment, social belongingness to a community, and regulating the psychological functioning of daily life were assessed using the MHC-SF test [157]. The three main categories of social, emotional, and psychological well-being were employed to gauge how well teenagers’ mental health was functioning.

The scale developed by [157] covered three dimensions: social well-being (5 items; α = 0.74), emotional well-being (3 items; α = 0.85), and psychological well-being (6 items; α = 0.84) and comprised 14 items [158]. The overall scale of MHC-SF reliability was α = 0.91. Respondents rate each item on a 7-point Likert scale, ranging from 1 (Very strongly disagree) to 7 (Very strongly agree). This scale possessed excellent construct validity [157]. In the present study, the three factor MHC-SF sub scales for both university and high school students: Emotional Well-Being (α = 0.951, 0.946; CR = 0.950, 0.945; AVE = 0.863, 0.852), Psychological Well-Being (α = 0.974, 0.969; CR = 0.973, 0.969; AVE = 0.858, 0.837), and Social Well-Being (α = 0.974, 0.971; CR = 0.974, 0.971; AVE = 0.882, 0.868), which showed high internal consistency, construct reliability, and discriminant validity respectively. The construct validity of the MHC-SF using CFA demonstrated an excellent fit to the current data for both university and high school students (χ^2^/df = 3.61, 3.70, TLI = 0.987, 0.985, CFI = 0.989, 0.988, RMSEA = 0.041, 0.043 (95% CI = 0.040, 0.041 to 0.062, 0.061), SRMR = 0.011, 0.013). Besides, the dependability and validity of the Amharic version have been proven in the Ethiopian context [6,143].

#### 3.4.7. The Scales over All Evaluation

The author(s) of the study conducted several analyses to evaluate the quality of the scales used in this research. The results showed that all estimated parameters for the scales were statistically significant, indicating that the scales effectively measured the intended constructs. Additionally, the composite reliability of the scales, which assesses the internal consistency or reliability of the items within each scale, ranged from 0.896 to 0.971. These values were higher than the recommended threshold of 0.60 proposed by [159]. This suggests that the scales demonstrated excellent internal consistency, implying that the items within each scale were highly reliable measures of the constructs.

Furthermore, the average variance extracted (AVE) for each scale, which measures the amount of variance captured by the scale’s items relative to measurement error, ranged from 0.685 to 0.868. These values exceeded the suggested threshold of 0.50 by [159]. This indicates that the scales exhibited good convergent validity and discriminant validity, effectively measuring the intended constructs and sharing a substantial amount of variance. Therefore, based on the analysis results, it can be concluded that the five scales used in the study demonstrated convergent validity, good discriminant validity, and satisfactory internal quality.

### 3.5. Data Analysis

To analyze the data, the study utilized two main statistical software packages, including SPSS version 29 and Smart PLS 4.0.1.3. Before commencing the analysis phase, several assumptions were carefully examined, including outliers, normality, linearity, multicollinearity, and singularity [156,160,161]. Once the necessary criteria were met, the data underwent analysis. It is worth noting that to avoid multicollinearity issues between main variables, it is recommended to maintain a correlation value below 0.90 [161]. In addition, to identify any potential issues with multi-collinearity in the data, the researchers used VIF (Variance Inflation Factor) and tolerance measures, as suggested by [6,19,156]. Additionally, the Harman single-factor test was used to assess the bias caused by common method variance based on the recommendation of [162]. Consequently, it was determined that the correlation value between binary variables in this study remained below this threshold. Furthermore, to confirm the normal distribution of the collected data should fall within the range of skewness (≤2) and kurtosis (≤4) values. Fortunately, the skewness and kurtosis values recorded in this study adhered to this criterion, thus satisfying the assumption of normality based on the assumption of [163]. In this study, the sample size is 300 or more, which supports the assumption of normality. The absolute values of skewness and kurtosis for the variables (Internet Addiction, Social Media Addiction, mindfulness, Social Capital, and Mental Health) are within the acceptable range for normal distribution (skewness ≤ 2, kurtosis ≤ 4) recommended by [163]. For the structural equation modelling analysis, various fit values deemed acceptable in the literature were adopted as the criteria for evaluation [161,164,165,166]. These criteria include a chi-square to degrees of freedom ratio (χ^2^/df) of less than 5, Tucker-Lewis index (TLI), comparative fit index (CFI), and incremental fit index (IFI) all above 0.90, root mean square error of approximation (RMSEA) and standardized root mean square residual (SRMR) below or equal to 0.08 [156]. Adhering to these criteria, the structural equation modelling analysis in this study was deemed appropriate for further interpretation.

In line with the recommended practices, confirmatory factor analysis (CFA) was conducted on the scales before testing the model [166]. Consequently, the measurement model was thoroughly tested, and subsequently, the formulated mediation model was examined. Consequently, CFA was conducted on the scales and the measurement and structural model were tested, then the formulated multi-mediation model was tested. Finally, to examine indirect effects, the researchers calculated 95% bias-corrected and accelerated confidence intervals using the bootstrap method with 5000 resamples.

### 3.6. Procedures of the Studies

#### 3.6.1. Data Collection System

The study involved participants from one large public university and five high schools which are found in the Amhara regional State of Ethiopia. University and high school students were chosen because they commonly use smartphones and actively engage with social networking sites for their learning. Due to students’ limited awareness of research importance and potential carelessness, an offline survey is the most suitable data collection system, which was conducted in our study to ensure reliable data. Participants were informed about the study’s objectives, and their participation was completely voluntary. Data collection took place over a three-month period from December to February during the 2023/2024 academic year.

#### 3.6.2. Adaptation, Translation, and Validation of the Measures

Cross-cultural validation was employed to adapt and validate instruments originally developed for other cultures [167]. This process ensures the suitability of the measures in a culturally diverse context like Ethiopia [167]. Various scales, such as the Internet Addiction Scale (IAS) [153], the Five Facet Mindfulness Questionnaire Short Form (FFMQ-SF) [126,127], the Personal Social Capital Scale (PSCS-8) [101]; the Bergen Social Media Addiction Scale (BSMAS) [150], and Keyes’ Mental Health Continuum-Short Form (MHC-SF) [152] were adapted and translated for use in the Ethiopian context. The recommended guidelines for cross-cultural validation were followed, including forward and back translation, synthesis, expert review, and testing validation [168]. The adapted instruments were then pretested and validated.

After validating the measures, the study aimed to conduct a multi-mediation analysis to explore the indirect effects of independent variables on a dependent variable through a series of mediators [169,170,171,172,173,174]. SMA is particularly useful when there is a sequential chain of mediators, where each mediator influences the next until the final mediator affects the dependent variable [149,169]. This analysis helps uncover the complex relationships and underlying mechanisms between variables. Confirmatory Factor Analysis (CFA) and structural equation modelling (SEM) was used to assess the measurement (factorial) validity and structural validity of the proposed models. Goodness-of-fit indices, such as normed chi-square, Tucker Lewis Index, Comparative Fit Index, Standardized Root Mean Residual, and Root Mean Squared Error of Approximation, were considered to evaluate the model fit [156]. CFA and SEM enhance the rigour and robustness of the analysis. The study identified mindfulness and social capital as influential factors mediating the relationship between internet addiction, social media addiction, and students’ mental health.

#### 3.6.3. Ethics of the Research

The data collection process for this research was conducted following the guidelines established by the American Psychological Association (APA). The study received approval ethical letter from the ethics committee at the first author’s university, Institute IRB, with the reference number 302/2023. All participants were adequately informed about the study, and the procedures were carried out in compliance with the Helsinki Declaration, including adherence to regulations such as 21 CFR 50 (Protection of Human Subjects) and 21 CFR 56 (Institutional Review Boards).

In this study, the students participating in the study neither received financial inducements nor any academic inducements such as academic credits. Participants were fully informed about the study’s purpose, procedures, and voluntary nature. They were assured of anonymity, and no identifying information or medical treatment was involved. Participants were aware of their right to withdraw from the study at any time. By initiating the survey, participants were considered to have read and accepted the informed consent. Throughout the research process, the researchers strictly adhered to the protocols, guidelines, and regulations outlined by the international research code of ethics. This approach ensured that students could engage in the research freely, fostering an environment of trust and integrity. By prioritizing voluntary participation, the study aimed to gather genuine insights while respecting the autonomy and well-being of the students involved.

## 4. Results

### 4.1. Demographic Data, Descriptive Statistics, Kurtosis, and Skewness

Table 1 provides the descriptive statistics, including the demographic detail data, mean, and standard deviation, for the different variables in the study.

The absolute values of kurtosis and skewness for the variables (social media addiction, internet addiction, social capital, mindfulness, and mental health) are within the acceptable range for normal distribution (kurtosis ≤ 4, skewness ≤ 2) recommended by [163,171]. In this study, the skewness values for social media addiction, internet addiction, social capital, mindfulness, and mental health for university and high school students were found to be (0.19, 0.09), (−1.06, −1.12), (−0.65, −0.60), (−0.98, −1.08), and (−0.60, −0.67), respectively. The kurtosis values for these constructs for university and high school students were found to be (−1.19, −1.22), (1.41, 1.68), (−0.02, −0.25), (0.81, 1.15), and (−0.053, 0.23), respectively.

#### 4.1.1. Multi-Collinearity Diagnostics

There are no issues with multicollinearity in our present study (see Table 2), as indicated by the Tolerance values of each predictor variables on criterion variable (mental health) being close to those in the model. Conversely, if the tolerance values are close to zero, it suggests a higher risk of multicollinearity [161]. To assess multicollinearity, the VIF statistic should ideally fall between 0 and 5, with lower numbers being more desirable, even approaching [161] . In our study, the VIF was below 5, indicating the absence of multicollinearity. Additionally, the tolerance limits for each independent variable were all greater than or equal to 0.01, further supporting the conclusion that our independent variables were free from multicollinearity issues, as measured by VIF and Tolerance.

Furthermore, we conducted the Harman single-factor test to investigate the presence of common method bias in our study for both university and high school students. The results revealed that all constructs exhibited a common method bias rate of only 31.58% and 30.50% for both university and high school students, which falls below (50%) the recommended fit requirements. Therefore, we concluded that, our study demonstrated the absence of multicollinearity issues through the favorable tolerance values and VIF scores. The low rate of common method bias further validated the reliability of our findings.

#### 4.1.2. Pearson Correlation Among the Study Constructs

For further analysis and steps establishing correlation among the constructs is the most important and preliminary. As a result, following the guidelines of [143,164,173], a correlation analysis was conducted to test the first hypothesis and determine the relationships between the independent factors (Social media addiction, internet addiction, social capital, mindfulness) and the criterion variable (mental health).

The results revealed a negative correlation between social media addiction (SMA) with the social capital (r = −0.20, *p* < 0.01; −0.18, *p* < 0.01), mindfulness (r = −0.33, *p* < 0.01; −0.22, *p* < 0.01), and mental health (−0.33, *p* < 0.01; −0.40, *p* < 0.01) on university and high school students, respectively. Besides, on both university and high school students, we found a negative correlation between internet addiction (IA) and social capital (r = −0.17, *p* < 0.01; −0.20, *p* < 0.01), mindfulness (r = −0.23, *p* < 0.01; −0.33, *p* < 0.01), and mental health (−0.42, *p* < 0.01; −0.34, *p* < 0.01), respectively. However, a positive significant correlation was found between mindfulness and social capital with mental health and among themselves (see Table 3).

Additionally, the gender (r = 0. 0.08, *p* < 0.01) and batch (r = 0.11, *p* < 0.01) showed a significant positive correlation with mental health of university students. However, gender had a negative correlation with social capital (r = −0.07, *p* < 0.01), and positively correlated with mental health (r = 0.071, *p* < 0.01) of high school students. Besides, high school students in terms of grade level had a positive correlation with mindfulness (r = 0.06, *p* < 0.05) and mental health (r = 0.11, *p* < 0.01) (see Table 3).

Furthermore, the absence of noteworthy correlations with socio-demographic factors suggests that these factors did not influence the relationship between the main constructs. Our analysis of Pearson correlation coefficients revealed no significant associations between the main constructs and variables such as gender, batch, and grade level (refer Table 3). As a result, we did not deem it necessary to further investigate the impact of socio-demographic factors on the main constructs.

### 4.2. Group Differences

To further analyze the differences between school types (university vs. high school), we conducted independent sample *t*-test (see Table 4), provides the actual results from the independent-samples *t*-test. In addition to this, Levine’s Test for Equality of Variances was computed. The significance (*p*-value) of Levene’s test is greater than 0.05 (the alpha level selected for the test) for all variables except mental health “Equal variance assumed” test was taken. For mental health, Equal variance not assumed test was taken. The results indicated that there were no statistically significant differences observed between school types: university and high school as demonstrated by Independent sample *t*-test for social media addiction, internet addiction, social capital, mindfulness, and mental health (t = 0.872, df = 263, *p* = 0.383; t = 0.886, df = 2631, *p* = 0.375; t = −0.554, df = 2631, *p* = 0.580; (t = −0.058, df = 2631, *p* = 0.954) and (t = −0.028, df = 2631, *p* = 0.977) respectively.

#### 4.2.1. Measurement and Structural Model

Measurement and structural models were then assessed. The measurement model consisted of five latent constructs and 15 indicators. The Personal Social Capital Scale (PSCS-8) had two indicators, the Five Facet Mindfulness Questionnaire Short Form (FFMQ-SF) was measured by five indicators, the Internet Addiction Scale (IAS) was measured by four indicators, the Bergen Social Media Addiction Scale (BSMAS) had one indicator, and Keyes’ Mental Health Continuum-Short Form (MHC-SF) had three indicators. The measurement model demonstrated a good fit based on the confirmatory factor analysis (CFA) results (see Appendix A from Appendix A for university students and from Appendix A for high school students).

After conducting an examination of the measurement model for the PSCS-8, FFMQ-SF, IAS, BSMAS, and MHC-SF, our next step involved assessing the overall constructs of the measurement model across various scales. This analysis encompassed model one (1) to model seven (7) for both university and high school students. The result demonstrated a good fit to the data for all models (model 1 to model 7) (see Table 5). These results indicate that the latent variables are accurately represented by their corresponding indicators. The structural model was then evaluated after confirmed measurement model. To test the structural model, we examined the fitness of indices of the latent constructs, and fitted to the data, yielding the acceptable fitness of indices. Finally, the full and the partial mediation model of structural models, namely Social Media Addiction → Social Capital and mindfulness → Mental health (Model 1-SMA); Social media addiction → Social Capital → Mental health (Model 2-SMA); Social media addiction → mindfulness → Mental health (Model 3-SMA); Internet addiction → Social Capital and mindfulness → Mental health (Model 4-IA); Internet addiction → Social Capital → Mental health (Model 5-IA); Internet addiction → mindfulness → Mental health (Model 6-IA) and Social media addiction and Internet addiction → Social Capital and mindfulness → Mental health (Model 7-SMA and IA) (see the details in Table 5). The findings revealed that, the measurement models and structural models (Model 1 to 7) demonstrated an acceptable fit to the data (see Table 5). All factor loadings were significant and ranged between 0.71 to 0.95, (*p* = 0.001), indicating that the indicators effectively captured the underlying latent variables. In summary, the measurement model and structural model for both university and high school students exhibited an acceptable fit to the data, indicating that the indicators accurately represented the latent constructs and the relationships between the constructs were well-supported by the data. The results of the study provide evidence for the reliability, convergent validity, discriminant validity, and construct validity of the social media addiction, internet addiction, social capital, mindfulness, and mental health constructs among undergraduate university and high school students. The structural model supports the proposed relationships between these constructs, confirming and proving the proposed hypotheses (see Table 5).

**Table 5 ijerph-22-00057-t005:** Confirmatory Factor Analysis of the Constructs Using the Measurement Model and the Structural Model.

**Models**	**Fitness of Indices**	**Confirmatory Factorial Analysis of the Variables**									
		**University Students**					**High School Students**				
		**χ^2^(df)**	**χ^2^/df**	**TLI**	**CFI**	**RMSEA**	**χ^2^(df)**	**χ^2^/df**	**TLI**	**CFI**	**RMSEA**
	Social media addiction	780 (9) *	6.70	0.911	0.912	0.085	888 (19) *	4.7	0.904	0.920	0.088
	Internet addiction	482 (113) *	4.27	0.973	0.978	0.053	658 (113) *	4.53	0.968	0.973	0.061
	Social Capital	318 (19) *	4.78	0.949	0.965	0.067	426 (20) *	4.43	0.946	0.963	0.061
	Mindfulness	701 (160) *	4.38	0.973	0.977	0.054	988(160) *	4.47	0.965	0.971	0.059
	Mental health	341 (74) *	4.61	0.987	0.989	0.041	422 (74) *	3.70	0.985	0.988	0.043
Model 1-SMA	Measurement Model	6686 (1025) *	5.01	0.912	0.920	0.069	10,836 (1065) *	5.17	0.901	0.908	0.079
	Structural model	7679 (111) *	5.21	0.901	0.907	0.073	10,836 (1065) *	5.17	0.901	0.908	0.079
Model 2-SMA	Measurement Model	1798 (342) *	5.30	0.963	0.968	0.060	2252 (342) *	4.58	0.959	0.963	0.062
	Structural model	1798 (342) *	5.30	0.962	0.968	0.061	2252 (342) *	4.58	0.959	0.963	0.062
Model 3-SMA	Measurement Model	5970 (729) *	4.94	0.927	0.931	0.058	7382 (729) *	4.83	0.903	0.909	0.079
	Structural model	5970 (729) *	4.94	0.927	0.931	0.058	7382 (729) *	4.83	0.903	0.909	0.079
Model 4-SMA	Measurement Model	7315 (1561) *	4.08	0.920	0.927	0.056	10,423 (1561) *	5.67	0.900	0.909	0.062
	Structural model	8408 (1633) *	5.15	0.911	0.915	0.059	11,910 (690) *	5.29	0.901	0.908	0.065
Model 5-IA	Measurement Model	2200 (1160) *	3.19	0.968	0.970	0.043	2749 (656) *	4.13	0.962	0.966	0.046
	Structural model	2200 (1160) *	3.19	0.968	0.970	0.043	2877 (690) *	3.17	0.962	0.964	0.046
Model 6-IA	Measurement Model	5970 (1209) *	4.94	0.925	0.929	0.061	7143 (1158) *	4.16	0.921	0.928	0.059
	Structural model	5970 (1209) *	4.94	0.927	0.931	0.058	8058 (1209) *	4.64	0.913	0.918	0.063
Model 7-MA and IA	Measurement Model	8708 (1910) *	4.56	0.916	0.923	0.055	12,211 (1910) *	5.39	0.901	0.906	0.061
	Structural model	9885 (1992) *	4.96	0.907	0.911	0.058	13,840 (1992) *	5.94	0.903	0.908	0.064
	Rule of Thumb		≤5	≥0.90		≥0.08		≤5	≥0.90		≥0.08

Note: * *p* < 0.001, χ^2^ = chi-squared, df = degrees of freedom, TLI = Tucker Lewis index, CFI = comparative fit index, RMSEA = root mean error square of approximation. Model 1-SMA: Social media addiction → Social Capital and mindfulness → Mental health (see Figure 2 and Figure 3). Model 2-SMA: Social media addiction → Social Capital → Mental health (see Section A.1 and Section B.1). Model 3-SMA: Social media addiction → mindfulness → Mental health (see Section A.2 and Section B.2). Model 4-IA: Internet addiction → Social Capital and mindfulness → Mental health (see Figure 4 and Figure 5). Model 5-IA: Internet addiction → Social Capital → Mental health (see Section A.3 and Section B.3). Model 6-IA: Internet addiction → mindfulness → Mental health (see Section A.4 and Section B.4). Model 7-SMA and IA: Social media addiction and Internet addiction → Social Capital and mindfulness → → Mental health (see Figure 6 and Figure 7).

**Figure 2 ijerph-22-00057-f002:**
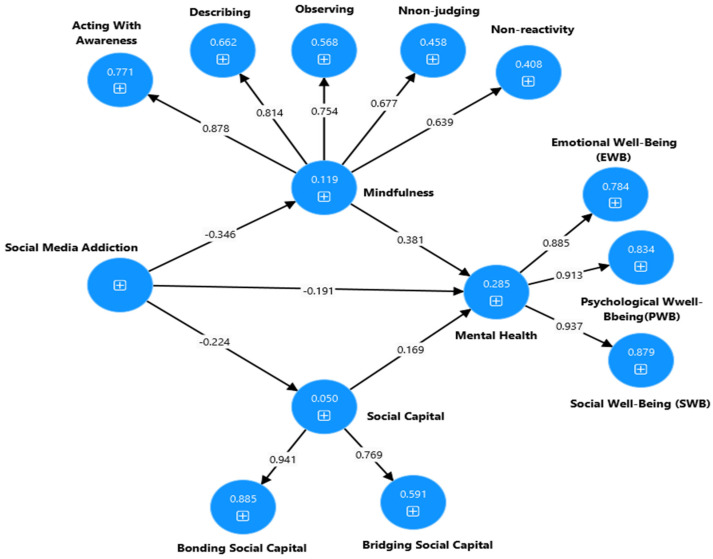
Output of the partial mediation model to explain the association between the social media addiction, social capital, mindfulness, and mental health of university students.

**Figure 3 ijerph-22-00057-f003:**
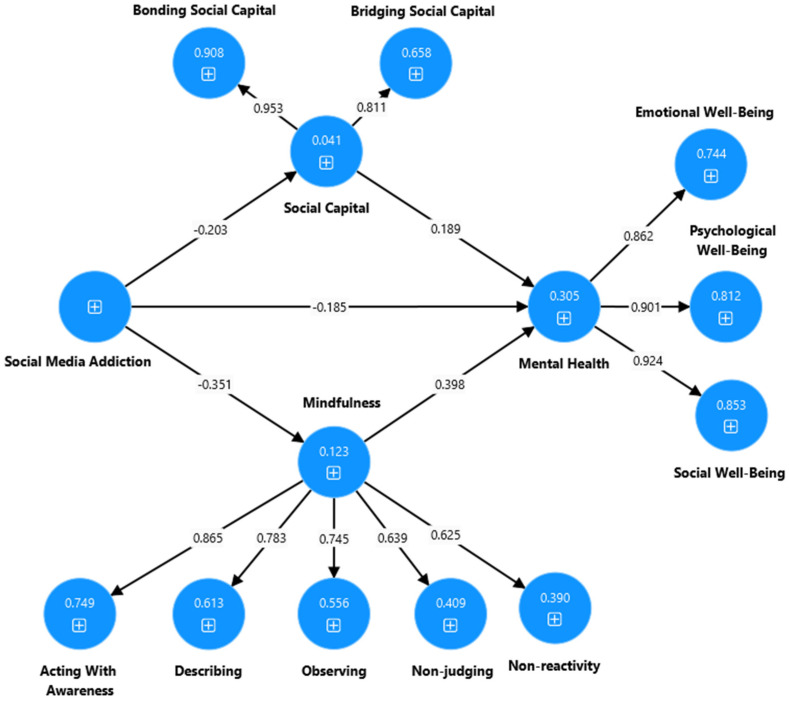
Output of the partial mediation model to explain the association between the social media addiction, social capital, mindfulness, and mental health of high school students.

**Figure 4 ijerph-22-00057-f004:**
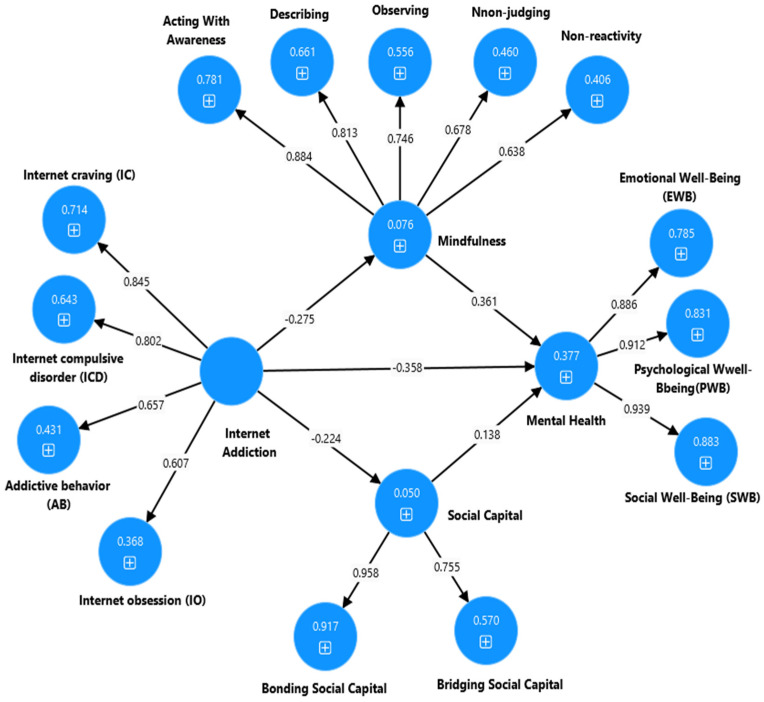
Output of the partial mediation model to explain the association between the Internet Addiction, social capital, mindfulness, and mental health of university students.

**Figure 5 ijerph-22-00057-f005:**
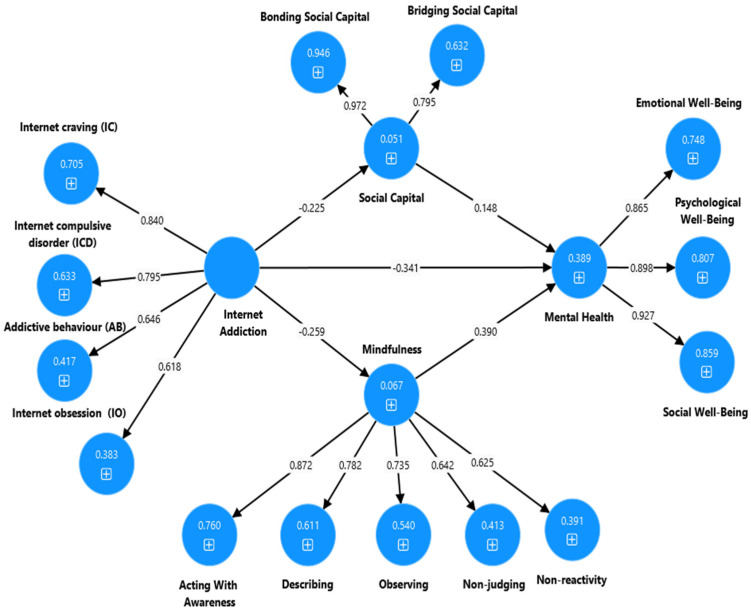
Output of the partial mediation model to explain the association between the internet addiction, social capital, mindfulness, and mental health of high school students.

**Figure 6 ijerph-22-00057-f006:**
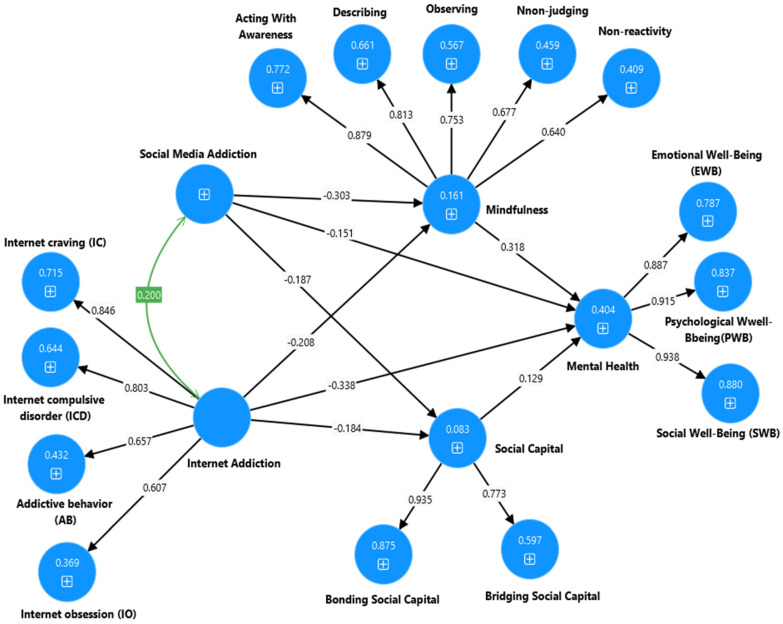
Output of complete multi-mediation model to explain the association between SMA, IA, social capital, mindfulness, and mental health of university students.

**Figure 7 ijerph-22-00057-f007:**
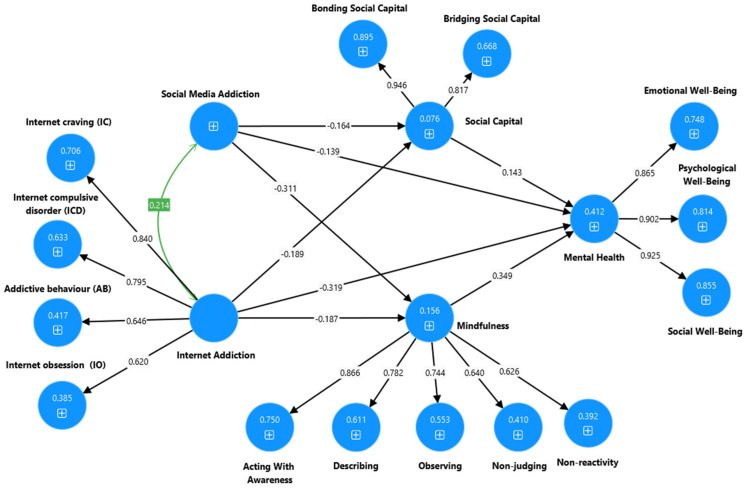
Output of Full multi-mediation model to explain the association between SMA, IA, social capital, mindfulness, and mental health of high school students.

#### 4.2.2. Partial (Single) Mediation Analysis

To further observe the mediating role of both social capital and mindfulness in the relationship between social media addiction (SMA) and mental health (MH) and internet addiction (IA) and mental health (MH), a partial (single) mediation analysis was first performed in Smart PLS software version 4.1.0.3 [172].

As we hypothesized (RH6a), the mediating effect of social capital between SMA and MH was significant (β = −0.065, [bootstrapped 95% CI: −0.095, −0.044], *p* = 0.001; β = −0.062, [bootstrapped 95% CI: −0.085, −0.041], *p* = 0.002) for both university and high school students, respectively. Besides, mindfulness was played a significant mediating effect on the relationship between SMA and MH for both university and high school students (β = −0.153, [bootstrapped 95% CI: −0.187, −0.117], *p* = 0.002; β = −0.163, [bootstrapped 95% CI: −0.1951, −0.130], *p* = 0.002), respectively.

In addition, social capital and mindfulness played a significant mediating role between SMA and MH for both university and high school students (β = −0.170, [bootstrapped 95% CI: −0.206, −0.133], *p* = 0.002; β = −0.178, [bootstrapped 95% CI: −0.213, −0.143], *p* = 0.002), respectively.

Furthermore, both social capital and mindfulness together partially mediated the relationship between IA and MH for both university and high school students (β = −0.134, [bootstrapped 95% CI: −0.173, −0.105], *p* = 0.001); β = −0.130, [bootstrapped 95% CI: −0.166, −0.096], *p* = 0.002), respectively. Besides, social capital played a significant mediating role between IA and MH (β = −0.055, [bootstrapped 95% CI: −0.082, −0.033], *p* = 0.001; (β = −0.059, [bootstrapped 95% CI: −0.084, −0.039], *p* = 0.001) for both university and high school students, respectively. Mindfulness also played a significant mediating effect on the relationship between IA and MH for both university and high school students (β = −0.107, [bootstrapped 95% CI: −0.140, −0.077], *p* = 0.002); β = −0.108, [bootstrapped 95% CI: −0.1401, −0.080], *p* = 0.001), respectively.

Our research hypothesis (RH6a) fully confirmed and after checking the single mediation analysis, we performed the full mediation or multi-mediation analysis for both university and high school students.

#### 4.2.3. Multi-Mediation Models

After conducting a thorough evaluation of construct reliability, measurement model (construct validity), and structural model tests using advanced methods, we proceeded to examine a multi-mediation models (MMM) using structural equation modeling (SEM) using PLS-SEM. This model serves as the analytical framework for systematically examining the relationships between independent and dependent variables and conducting hypothesis testing [173]. To test a multiple mediation model following the recommendation of the authors [173], we followed the steps of first running a set of separate single mediation analyses, one for each proposed single mediator separately, followed by testing the indirect effects (i.e., each specific and total indirect effect) and the direct effect between the independent variables and the dependent variable computed.

The structural model provides several statistical metrics, including coefficient of determination (R^2^), predictive relevance, and direct and indirect path coefficients, which are important for our analysis [174].

Our research objectives state that students believe social media and internet addiction have a negative influence on their mental health, with mindfulness and social capital as played a mediating role. To assess the significance of the predictive relationships, we employed corrected bias bootstrapping in the Smart PLS software [172]. This method helps determine the directionality (positive or negative), magnitude, and significance of the independent variables’ (social media addiction, internet addiction, social capital and mindfulness) influence on the dependent variables (mental health) [172,173,174].

Table 6 showed the direct effect analysis for each path. The proposed research hypotheses (RH1to RH4) focused on “Do Internet Addiction (IA) and Social Media Addiction (SMA), have a direct negative effect on social capital, mindfulness, and mental health (MH) among both university and high school students? Results indicated that SMA and IA had a negative significant indirect effect on university students MH through social capital and mindfulness, supporting our research hypothesis (RH6a) (β = −0.121 [bootstrapped 95% CI: −0.154 to −0.096], *p* = 0.001) and β = −0.090 [bootstrapped 95% CI: −0.119 to −0.066], *p* = 0.001), respectively (See Figure 2). University students also perceived both SMA and IA to exert a negative direct effect on mindfulness (β = −0.303, [bootstrapped 95% CI: −0.354, −0.251], *p* = 0.002; β = −0.208, [bootstrapped 95% CI: −0.261, −0.146], *p* = 0.002), social capital (β = −0.188, [bootstrapped 95% CI: −0.244, −0.129], *p* = 0.002; β = −0.184, [bootstrapped 95% CI: −0.254, −0.111], *p* = 0.002) and MH (β = −0.151, [bootstrapped 95% CI: −0.204, −0.102], *p* = 0.002; β = −0.338, [bootstrapped 95% CI: −0.390, −0.283], *p* = 0.003), respectively. Furthermore, university students believe that social capital (β = 0.129, [bootstrapped 95% CI: 0.067 to 0.194], *p* = 0.001) and mindfulness (β = 0.318, [bootstrapped 95% CI: 0.250 to 0.385], *p* = 0.002) positively affects both their MH. Therefore, RH1 to RH4 are fully supported, as displayed in Table 6. Besides, in Figure 3, we observed a tested model aimed at investigating whether social media addiction (SMA) use independently or partially influences university students’ mental health. This investigation considered the mediating factors of mindfulness and social capital, both individually and in combination. The results revealed several significant relationships. Firstly, we found that SMA had a direct negative effect on social capital, mindfulness, and mental health in the partial model with the mediating factors of social capital, and mindfulness as indicated in Table 6 and Table 7. SMA indirectly affects university students’ mental health separately through social capital and mindfulness. This suggests that SMA without internet addiction (IA) in a separate model predicts that students with low SMA engagement tend to exhibit greater social capital, mindfulness, and mental health. This suggests that SMA without IA in separate model predicted students who possess a low engagement of SMA tend to exhibit greater social capital, mindfulness, and mental health. Secondly, IA was found to have a negative direct effect on university students’ mental health through social capital and mindfulness. This implies that students with high internet addiction are more likely to demonstrate lower social capital, mindfulness, and mental health, and vice versa (see Table 6). Thirdly, IA showed a negative direct effect on social capital and MH through social capital as well as through mindfulness, suggesting that students who have better social capital resources and mindfulness skills tend to reduce IA and boost students’ mental health. Our data thus answered RH3 and RH4.

Regarding high school students, our findings found that SMA and IA had a negative significant indirect effect on students MH through social capital and mindfulness, β = −0.132, [bootstrapped 95% CI: −0.162, −0.104], *p* = 0.002; β = −0.092, [bootstrapped 95% CI: −0.121, −0.068], *p* = 0.001, respectively, supporting our research question (RH6a) (see Table 6). We also found that SMA and IA had a direct significant effect on the three constructs namely; mindfulness (β = −0.310, [bootstrapped 95% CI: −0.358, −0.261], *p* = 0.003; β = −0.187, [bootstrapped 95% CI: −0.245, −0.1344], *p* = 0.001); social capital (β = −0.164, [bootstrapped 95% CI: −0.211, −0.109], *p* = 0.003; β = −0.189, [bootstrapped 95% CI: −0.254, −0.132], *p* = 0.001); and Mental Health (β = −0.139, [bootstrapped 95% CI: −0.186, −0.090], *p* = 0.002; β = −0.319, [bootstrapped 95% CI: −0.375, −0.266], *p* = 0.002), respectively. Furthermore, university students believe that social capital (β = 0.143, [bootstrapped 95% CI: 0.086 to 0.195], *p* = 0.002) and mindfulness (β = 0.349, [bootstrapped 95% CI: 0.289 to 0.419], *p* = 0.001) positively predicts both their MH. Therefore, RH1 to RH4 are fully supported, as displayed in Table 6. In our multi- mediation model, the potential predictive power of social media addiction and internet addiction using R^2^ values for social capital (0.083 (8.3%), 0.076 (7.6%), for mindfulness (0.161 (16.1%), 0.156 (15.6%)) and for mental health (0.404 (40.4%), 0.412 (41.2%)) for both university and high school students, respectively. The comprehensive set of statistical techniques of all measurement and structural model outcomes can be found in Table 5, and the structural model is visually represented in Figure 2, Figure 3, Figure 4, Figure 5, Figure 6 and Figure 7, Section A.1, Section A.2, Section A.3, Section A.4, Section B.1, Section B.2, Section B.3 and Section B.4.

## 5. Discussion

Consistent with the results of various previous studies, this research confirmed the negative relationships between social media addiction (SMA) and internet addiction (IA) with mindfulness, social capital, and mental health (MH) for both university and high school students [11,12,50,83,90,94,99,100,111,112,113,114,115,116,136,137]. Additionally, this study built a seria mediation or multi-mediation effect model. It revealed that when taking mindfulness and social capital as mediating variables, both play multiple roles in the relationship with SMA, IA, and MH: First, SMA has a direct negative effect on mindfulness, social capital, and mental health for both university and high school students, Second, IA has a direct negative effect on mindfulness, social capital, and mental health for both university and high school students, Third, mindfulness and social capital also had a positive direct effect on both university and high school students mental health. Fourth, through mindfulness and social capital, SMA and IA also indirectly influence the mental health of university and high school students. A high level of mindfulness and social capital resources helps reduce the risk of SMA and IA for both university and high school students. Fifth, mindfulness partially mediates the relationships between SMA and mental health as well as IA and mental health for both university and high school students. Sixth, social capital also partially plays a mediating role in SMA and mental health, as well as IA and mental health for both university and high school students. The measurement and structural model (models 1 to 7) confirmed the model fitness and applicability of the proposed multi-mediation models. Overall, the positive psychology constructs and social cognitive constructs namely mindfulness and social capital imposes a multiple direct and mediating effect on mental health. In other words, even not considering the direct effects, mindfulness and social capital not only influences directly but also play a mediating role among SMA, IA, and mental health for both university and high school students. The findings suggest that mindfulness and social capital play an important role in buffering and mediating the impacts of SMA and IA on the students MH. From a positive psychology lens, these results offer valuable insights for designing SMA and IA interventions targeting university and high school students.

Specifically, consistent with several previous studies regarding the relationship among the constructs [50,81], the results revealed a significant negative correlation between SMA with mindfulness [135], social capital [137], and mental health [66,67,68,78,105]. Similarly, IA and mindfulness [19,123,126,128], social capital [121], and mental health [6]. This highlights the detrimental impact of SMA and IA on these important psychological factors [19,42,43,44]. However, it is worth noting that when used appropriately, SMA and IA can have positive impacts, such as facilitating virtual interactions and supporting academic achievement [6,125,143].

The study’s key contribution is the examination of the mediating roles of mindfulness and social capital in the relationship between SMA and MH, IA and MH separately and together. The results revealed that both mindfulness and social capital partly (see Figure 1, Figure 2, Figure 3 and Figure 4, Section A.1, Section A.2, Section A.3, Section A.4, Section B.1, Section B.2, Section B.3 and Section B.4) and fully mediated (see Figure 5 and Figure 6) the negative relationship between SMA and MH, and IA and MH. Our findings align with research on the detrimental effects of SMA and IA on mental health [6,42,43,44,143], and the protective role of mindfulness and social capital [6,143] and social capital [84,86]. Specifically, the study confirmed that higher social capital and mindfulness lead to lower SMA and IA, and better mental health among both university and high school students, consistent with the protective role of mindfulness and social capital in mediating adverse experiences and boosting adolescents’ mental health [81,82,83,84,85,86]. Additionally, mindfulness and social capital was found to partially mediate the relationship between SMA and MH, as well as IA and MH, highlighting the importance of fostering social capital resources and mindfulness intervention to mitigate the negative impacts of SMA, IA, to foster mental health of young adolescents and to keep healthy digital system.

This study also investigated the direct effects of SMA, IA, mindfulness, and social capital on mental health. Several studies consistently with our study found that SMA and IA have negative impacts on mental health, positive psychological variables, academic performance, mindfulness, moral potency, sleep patterns, and overall well-being [19,42,43,44]. Interestingly in this study, SMA and IA was found to be a negative predictor of social capital, mindfulness, and MH. The results confirmed that social capital and mindfulness are a significant positive predictor of MH, supporting the idea that social capital and mindfulness are positively related to MH [18,47,48,49,50,51,92,129]. However, the complex relationships and influence of each predictor on mental health of high school and university students require further investigation [114]. For university and high school students, mindfulness [6] and social capital [36,65,84,86] are critical important positive factors that reduce SMA and IA. The relationship between mindfulness, IA, and SMA has been identified, and mindfulness and social capital can be helpful to decreasing SMA and IA [9]. Taking into consideration university and high school students, this study found that reducing Social Media Addiction (SMA) and internet addiction (IA) through high levels of social capital and mindfulness boosts their mental health status. These findings emphasize the need for interventions that address both SMA and IA as potential protecting factors both mindfulness and social capital played a mediating role in enhancing MH among university and high school students. The study’s contributions include addressing conceptual, methodological, and contextual gaps by testing a multi-mediation analysis using Structural Equation Modeling (SEM) related to the scarcity of this associational research in Ethiopian and global contexts, as well as providing new insights into the underlying mechanisms and mediators of the relationships between SMA, IA, social capital, mindfulness, and MH. The cultural and institutional context of Ethiopia adds unique perspectives to the existing literature from developing country perspectives.

Overall, the research on the connections among internet addiction (IA), social media addiction (SMA), social capital, mindfulness, and mental health addresses a pressing issue in today’s digital landscape. Based on the major findings, we suggest the following recommendations for educators, researchers, and policymakers.

General recommendation: Preventive Strategies for Internet and Social Media Addiction

Several studies have found that the best preventive strategies for internet and social media addiction, based on empirical research findings, are the following. These strategies will help reduce internet and social media addiction and improve the mental health status of adolescents.

Schools as Prevention Hubs: Schools should act as information and prevention hubs, aiding in early identification of problematic internet use and collaborating with mental health services to support students [175].

Mental Health Literacy: Enhance mental health literacy through systemic changes in education systems and evidence-based parental education to raise awareness of the positive and negative impacts of technology [175].

Psychoeducation and Upskilling: Address psychosocial and communication issues, focusing on self-regulation, content-related concerns like sexting, and context-related issues such as cyberbullying and body image problems [175].

Emotion Regulation Strategies: Utilize sensory methods, imaginative techniques, and mindfulness-based methods to improve psychophysiological emotion regulation [176].

Behavioral Psychology and Neurobiological Etiology Models: Integrate neurobiological insights with behavioral understanding to address Internet Use Disorders effectively, drawing parallels with substance-related addictions [176].

The PROTECT Etiology Model: Strengthen targeted behavioral, cognitive, and emotional therapies to combat Internet Use Disorders driven by gratification and compensation mechanisms [176].

Health Policy Strategy: Develop robust health policy strategy prioritizing research on specific forms of problematic internet use within addiction studies to establish evidence-based harm-reduction policies [177].

Cognitive Behavioral Model (CBM): Implementing CBM to target and modify maladaptive cognitions associated with internet addiction (IA) is an effective strategy for addressing cognitive distortions linked to IA [178].

Recommendations for Researchers

Conduct longitudinal studies to evaluate the long-term efficacy of mindfulness and social capital interventions in reducing social media and internet addiction.Investigate how cultural variations influence the associations between social media addiction, internet addiction, mindfulness, social capital, and mental health [92,127,130,131,143].

Recommendations for Educators

Incorporate mindfulness practices into the curriculum to enhance attention control, emotional regulation, and stress management skills [19].

Recommendations for Policy Makers

Develop digital wellness policies promoting guidelines for addressing social media and internet addiction among students [175,176,177,178,179].Design training programs for educators to address digital addiction and mental health issues in educational settings [177].Enhance digital literacy through programs encouraging appropriate internet and social media usage.Introduce school-based interventions focusing on resilience, coping strategies, and social skills to counteract negative effects of social media and internet addiction.

## 6. Conclusions

The findings of this study revealed that SMA and IA among university and high school students exhibited widespread complex relationships with social capital, mindfulness, and intellectual health (MH). This study confirms the negative effects of SMA and IA on students’ mental health and highlights the intricate dynamics through which mindfulness and social capital mitigate these adverse effects. The study demonstrates that both SMA and IA exert direct negative influences on mindfulness, social capital, and mental health among students. This direct effect underscores the pervasive detrimental impact of excessive social media and internet use. Moreover, the positive direct effects of mindfulness and social capital on mental health indicate that these constructs serve as critical buffers against the negative impacts of SMA and IA. Furthermore, the research delineates the indirect pathways through which SMA and IA affect mental health via mindfulness and social capital. By partially mediating these relationships, mindfulness and social capital not only buffer the direct negative effects but also facilitate a healthier mental state among students. This dual role of direct influence and mediation underscores the importance of fostering both mindfulness and social capital as protective factors in interventions targeting SMA and IA. The finding of this compressive study recommends the need for multifaceted interventions that address not only IA and SMA but also encouraging mindfulness and social capital development, such as promoting digital literacy, fostering mindfulness practices, and enhancing social support networks to manage their social media and internet usage and improve their mental health.

## 7. Limitation and Future Research Direction

In recent years, researchers in the fields of education, psychology, and health sciences have studied the crucial links between social media addiction (SMA), internet addiction (IA), social capital, mindfulness, and mental health in young populations. These studies have highlighted the potential mediating roles of mindfulness and social capital in this relationship and underscored their practical relevance for both general education (secondary schools) and higher education (universities and colleges). First, key findings from this body of research suggest that SMA and IA negatively predict social capital, mindfulness, and mental health among both university and high school students. That is, as students’ levels of mindfulness and social capital improve, their SMA and IA decrease and their mental state increases. These two positive psychological resources—mindfulness and social capital—show promise as interventions to improve student mental health in both academic settings. Second, understanding the factors that influence mental health is a critical global agenda because organizational success, productivity, and the overall quality of life of individuals are linked to the mental health status of the productive population. As a long-term goal, practitioners and researchers are encouraged to broadly focus on positive psychology theories (mindfulness, positive psychological capital, emotional intelligence) and Albert Bandura’s social cognitive theory (social capital) to improve students’ mental health using the latest scientific findings and sophisticated research methods. Third, an important limitation of this study is its dependence on variables assessed through scale scores, which may not directly align with the clinical diagnosis of social media addiction or internet addiction, commonly approached syndromally. Future studies should address this limitation by incorporating diagnostic criteria to enhance the clinical relevance and applicability of findings in real-world settings.

Fourth, while this research examined mental health in an educational context, the models developed and tested can be used in a variety of other settings, such as clinical, marketing, and other organizations, to address issues associated with SMA and IA across diverse populations.

Fifth, to further strengthen the evidence, future research should consider engaging not only students but also teachers and other health professionals to gain diverse perspectives on the links between SMA, IA, social capital, mindfulness, and mental health. Additionally, intervention studies that provide university and high school students with training in mindfulness strategies and sources of social capital could offer valuable insights into improving their mental health, using social media and the internet in healthy ways, and improving mindfulness intervention skills. Six, while this study used the common method Bises (CMB) using the Herman single-factor solution, VIF/Tolerance and reported reliable findings, the use of a self-report measure and a single data source increases the potential for single method bias. Finally, while this study utilizes mediation analysis to explore potential indirect relationships among variables, it is important to acknowledge a significant limitation due to the inability to infer causality. Mediation analysis can indicate associations and suggest pathways, but it does not establish definitive cause-and-effect relationships. Therefore, we recommend that future studies conduct experimental and quasi-experimental designs.

Moreover, this study is limited due to cross-sectional design, it could not show a temporal relationship and the scope of addiction measures. While the study focuses on general internet and social media addiction, it does not extend to clinical diagnoses of social media addiction disorder or internet addiction disorder. This may limit the generalizability of the findings to more severe cases and the broader spectrum of addiction behaviors. Future research could benefit from including these clinically defined disorders to better understand their impact on students’ mental health. Future research should explore meta-analytic, longitudinal, and experimental approaches to address these methodological limitations and further validate the relationships explored in this important area of inquiry.

## Figures and Tables

**Figure 1 ijerph-22-00057-f001:**
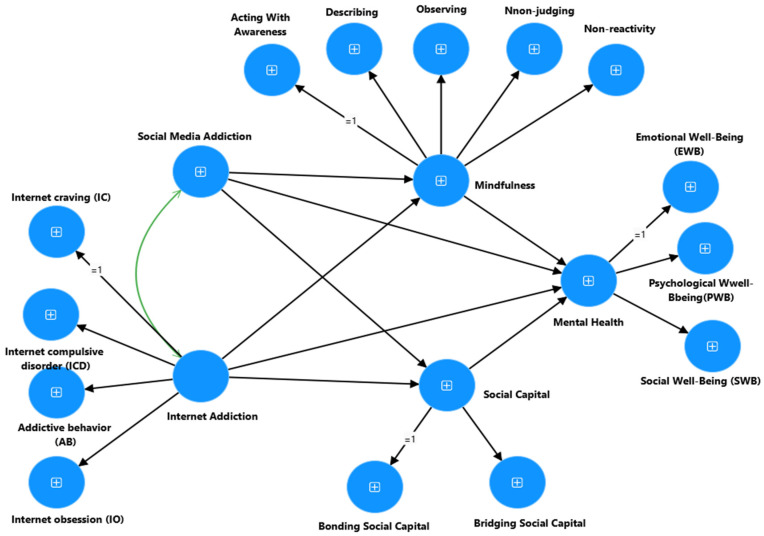
Proposed Full multi-mediation model to explain the association between social media addiction, internet addiction, social capital, mindfulness, and mental health. Note. The figure illustrates the proposed associations among the constructs in the theoretical model.

**Table 1 ijerph-22-00057-t001:** Descriptive statistics, kurtosis, and skewness of high school and university students, respectively.

Variables	Min		Max		Mean		Std. Dev		Skewness		Kurtosis	
	Univ.	High sc.	Univ.	High sc.	Univ.	High sc.	Univ.	High sc.	Univ.	High sc.	Univ.	High sc.
Interent caraving	5.00	5.00	28.00	28.00	20.00	20.01	5.06	4.98	−1.09	−1.07	0.60	0.58
Internet compulsive disorder	4.00	4.00	22.00	22.00	14.75	14.81	4.27	4.20	−0.76	−0.78	−0.18	−0.11
Addictive behaviour	4.00	4.00	20.00	20.00	15.29	15.07	4.07	4.04	−0.88	−0.82	0.07	0.02
Internet obsession	4.00	4.00	20.00	20.00	14.85	15.03	4.65	4.42	−0.77	−0.88	−0.47	−0.14
Acting with awareness	4.00	4.00	24.00	24.00	15.88	16.07	5.47	5.32	−0.79	−0.84	−0.13	0.04
Describing	4.00	4.00	24.00	24.00	15.91	16.05	5.54	5.33	−0.80	−0.83	−0.18	−0.01
Observing	4.00	4.00	24.00	24.00	13.89	14.13	5.41	5.33	−0.26	−0.29	−0.79	−0.73
Non-judging	4.00	4.00	24.00	24.00	12.68	12.77	4.99	4.81	−0.16	−0.22	−0.75	−0.67
Non-reactivity	4.00	4.00	25.00	25.00	15.37	15.18	5.37	5.18	−0.61	−0.59	−0.26	−0.17
Emotional well-being	3.00	3.00	21.00	21.00	11.04	11.00	4.57	4.38	−0.23	−0.25	−0.71	−0.62
Psyhological well-being	6.00	6.00	42.00	42.00	23.41	23.43	8.96	8.54	−0.49	−0.53	−0.43	−0.26
Social well-being	5.00	5.00	35.00	35.00	19.96	20.02	7.62	7.27	−0.60	−0.66	−0.44	−0.27
Bonding	4.00	4.00	24.00	24.00	15.88	15.66	5.08	5.37	−0.83	−0.77	−0.03	−0.30
Brdging	4.00	4.00	24.00	24.00	14.81	14.69	5.42	5.37	−0.41	−0.35	−0.65	−0.68
Social media addiction	3.00	6.00	36.00	36.00	16.35	16.62	8.020	7.87	0.19	0.09	−1.19	−1.22
SocialCapital	8.00	8.00	48.00	48.00	30.70	30.36	9.63	9.95	−0.65	−0.60	−0.02	−0.25
Mindfulness	20.00	20.00	120.00	120.00	73.75	74.20	21.41	20.37	−0.98	−1.08	0.81	1.15
Mental health	14.00	14.00	98.00	98.00	54.42	54.46	19.85	18.76	−0.60	−0.67	−0.053	0.23
Intrerent Addiction	17.00	17.00	90.00	90.00	64.90	64.91	14.23	13.87	−1.06	−1.12	1.41	1.68
Demographic Data of the participants												
High School				University								
Sex	Male	873	59.3%	Sex	Male	689	59.4%					
	Female	600	40.7%		Female	471	40.6%					
Name of high schools	Hotie	471	32%	Batch	Freshman	635	54.7					
	Memihir Akalewold	375	25.5%		Sophomre	403	34.7					
	Kidame Gebeya	275	18.7%		Senior	122	10.5					
	Nigus M.	199	13.5%									
	Tita	153	10.4%									
Grade Level	9	507	34.4%									
	10	502	4.1%									
	11	464	31.4									

Notes: High Sc. = High School students; Max = maximum; Min = minimum; Std. Dev = standard deviation; Univ = University Students.

**Table 2 ijerph-22-00057-t002:** VIF and Tolerance of multi-collinearity statistics on students’ mental health.

Model		Standardized Coefficients	t	Sig.	Collinearity Statistics	
		Beta			VIF	Tolerance
University students	Internet Addiction	−0.30	−12.29	0.000	1.08	0.92
	Social Meda Addiction	−0.15	−5.93	0.000	1.15	0.87
	Social Capital	0.10	3.86	0.000	1.23	0.81
	Mindfulness	0.31	11.53	0.000	1.35	0.74
High school students	Internet Addiction	−0.28	−12.79	0.000	1.08	0.92
	Social Media Addiction	−0.14	−6.47	0.000	1.15	0.87
	Social Capital	0.12	5.10	0.000	1.24	0.81
	Mindfulness	0.33	13.71	0.000	1.34	0.74

**Table 3 ijerph-22-00057-t003:** Pearson correlations (r) among the socio-demographic factors and the study variables.

Variables	University Students (*n* = 1160)						
	1	2	3	4	5	6	7
1. Gender	1						
2. Batch	0.04	1					
3. Internet Addiction	−0.04	−0.01	1				
4. Social Meda Addiction	−0.02	−0.06	0.20 **	1			
5. Social Capital	−0.03	0.01	−0.17 **	−0.20 **	1		
6. Mindfulness	−0.01	0.02	−0.23 **	−0.33 **	0.43 **	1	
7. Mental health	0.06	0.05	−0.42 **	−0.33 **	0.32 **	0.48 **	1
	High school students (*n* = 1473)						
Variables	1	2	3	4	5	6	7
1. Gender	1						
2. Grade level	0.04	1					
3. Internet Addiction	−0.03	−0.05	1				
4. Social Meda Addiction	−0.02	−0.04	0.22 **	1			
5. Social Capital	−0.04	−0.03	−0.20 **	−0.18 **	1		
6. Mindfulness	−0.03	0.06	−0.33 **	−0.22 **	0.43 **	1	
7. Mental health	0.06	0.05	−0.34 **	−0.40 **	0.34 **	0.49 **	1

Note: **. Correlation is significant at the 0.01 level (2-tailed).

**Table 4 ijerph-22-00057-t004:** T-test results for equality of mean scores (Levene’s Test for Equality of Variances) by school type on main variables (N = 2633, df = 2631).

Variables	Levene’s Test for Equality of Variances	F	Sig.	t	Sig. (2-Tailed)	University (N = 1160)		High School (N = 1473)	
						Mean	Std.	Mean	Std.
SMA	Equal variances assumed	0.565	0.452	−0.872	0.383	16.35	8.02	16.63	7.88
	Equal variances not assumed			−0.871	0.384				
SC	Equal variances assumed	2.439	0.118	0.886	0.375	30.70	9.64	30.35	9.95
	Equal variances not assumed			0.890	0.374				
Mind.	Equal variances assumed	2.776	0.096	−0.554	0.580	73.7517	21.41	20.37	20.37
	Equal variances not assumed			−0.551	0.582				
MH	Equal variances assumed	5.059	0.025	−0.059	0.953	54.42	19.85	54.46	18.76
	Equal variances not assumed			−0.058	0.954				
IA	Equal variances assumed	0.792	0.374	−0.028	0.977	64.90	14.23	64.91	13.87
	Equal variances not assumed			−0.028	0.977				

Note: IA = Internet Addiction, Mind = Mindfulness, MH = Mental Health, SC = Social Capital, SMA = Social Media Addiction. Interpretation of the independent-samples *t*-test in this study: If the Sig. value is larger than 0.05 (e.g., 0.07, 0.10) and if the significance level of Levene’s test is *p* = 0.05 or less (e.g., 0.01, 0.001), you should use the first line in the table, which refers to equal variances assumed and equal variances not assumed, respectively. Also, if the value in the Sig. (2-tailed) column is equal or less than 0.05 (e.g., 0.03, 0.01, 0.001) , then there is a significant difference in the mean scores on the study variable for each of the two groups and if the *p* value (Sig. 2-tailed) is above 0.05 (e.g., 0.06, 0.10), there is no significant difference between the two groups.

**Table 6 ijerph-22-00057-t006:** The direct and indirect effects of independent variables on students’ mental health using a 95% biased corrected confidence interval.

Standardized Direct Effect									
Predictors	Outcome Variables	Bootstrap 95% CI for University Students (N = 1160)				Bootstrap 95% CI for High School Students (N = 1473)			
		Beta	LBC	UBC	*p*-Value	Beta	LBC	UBC	*p*-Value
Social media addiction	Mindfulness	−0.303	−0.354	−0.247	0.002	−0.310	−0.358	−0.261	0.003
Social media addiction	Social Capital	−0.188	−0.244	−0.128	0.002	−0.164	−0.211	−0.109	0.003
Social media addiction	Mental health	−0.151	−0.197	−0.096	0.003	−0.139	−0.186	−0.090	0.002
Internet addiction	Mindfulness	−0.208	−0.267	−0.146	0.002	−0.187	−0.245	−0.134	0.001
Internet addiction	Social Capital	−0.184	−0.258	−0.108	0.002	−0.189	−0.254	−0.132	0.001
Internet addiction	Mental health	−0.318	−0.253	−0.394	0.001	−0.319	−0.375	−0.266	0.002
Mindfulness	Mental health	0.318	0.253	0.394	0.001	0.349	0.289	0.419	0.001
Social Capital	Mental health	0.129	0.059	0.193	0.004	0.143	0.086	0.195	0.002
Partial Mediation: Direct Effect of SMA on MH through Mindfulness and Social Capital									
Social media addiction	Mindfulness	−0.346	−0.396	−0.290	0.002	−0.351	−0.391	−0.301	0.004
Social media addiction	Social Capital	−0.224	−0.283	−0.162	0.002	−0.203	−0.253	−0.143	0.003
Social media addiction	Mental health	−0.191	−0.243	−0.136	0.003	−0.185	−0.232	−0.137	0.001
Mindfulness	Mental health	0.381	0.310	0.455	0.002	0.398	0.335	0.469	0.001
Social Capital	Mental health	0.169	0.101	0.232	0.002	0.189	0.136	0.251	0.002
Partial Mediation: Direct Effect of SMA on MH through Mindfulness									
Social media addiction	Mindfulness	−0.342	−0.392	−0.286	0.002	−0.348	−0.388	−0.298	0.004
Social media addiction	Mental health	−0.199	−0.249	−0.143	0.002	−0.190	−0.236	−0.140	0.002
Mindfulness	Mental health	0.447	0.387	0.505	0.002	0.469	0.418	0.529	0.001
Partial Mediation: Direct Effect of SMA on MH through Social Capital									
Social media addiction	Social Capital	−0.208	−0.268	−0.148	0.002	−0.190	−0.241	−0.131	0.003
Social media addiction	Mental health	−0.287	−0.342	−0.233	0.002	−0.292	−0.340	−0.245	0.001
Social Capital	Mental health	−0.312	−0.259	−0.371	0.001	0.325	0.274	0.377	0.002
Partial Mediation: Direct Effect of IA on MH through Mindfulness and Social Capital									
Internet addiction	Mindfulness	−0.275	−0.338	−0.209	0.002	−0.259	−0.320	−0.203	0.001
Internet addiction	Social Capital	−0.224	−0.310	−0.144	0.001	−0.225	−0.292	−0.160	0.001
Internet addiction	Mental health	−0.358	−0.411	−0.304	0.002	−0.341	−0.395	−0.3292	0.002
Mindfulness	Mental health	0.361	0.287	0.429	0.002	0.390	0.332	0.460	0.001
Social Capital	Mental health	0.138	0.073	0.204	0.002	0.148	0.092	0.204	0.002
Partial Mediation: Direct Effect of IA on MH through Mindfulness									
Internet addiction	Mindfulness	−0.254	−0.311	−0.190	0.002	−0.238	−0.296	−0.183	0.001
Internet addiction	Mental health	−0.362	−0.412	−0.307	0.002	−0.348	−0.401	−0.300	0.002
Mindfulness	Mental health	0.423	0.362	0.478	0.002	0.452	0.405	0.515	0.001
Partial Mediation: Direct Effect of IA on MH through Social Capital									
Internet addiction	Social Capital	−0.188	−0.265	−0.116	0.002	−0.197	−0.259	−0.141	0.001
Internet addiction	Mental health	−0.415	−0.465	−0.362	0.002	−0.397	−0.445	−0.351	0.001
Social Capital	Mental health	0.290	0.233	0.350	0.002	0.300	0.247	0.350	0.002

Note: The model description of predictors, mediators, and outcome variables. CI = Confidence interval; LBC = Lower bound, UBC = Upper bound.

**Table 7 ijerph-22-00057-t007:** Bootstrapping standardized indirect effect using 95% biased corrected confidence interval predicting Students both university and high school students.

Standardized Indirect Effect									
Models		Bootstrap 95% CI for University Students (N = 1160)				Bootstrap 95% CI for High School Students (N = 1473)			
		Beta	LBC	UBC	*p*-Value	Beta	LBC	UBC	*p*-Value
Model 1-SMA		−0.170	−0.206	−0.133	0.002	−0.178	−0.213	−0.143	0.002
Model 2-SMA		−0.065	−0.095	−0.044	0.001	−0.062	−0.085	−0.041	0.002
Model 3-SMA		−0.153	−0.187	−0.117	0.002	−0.163	−0.195	−0.130	0.002
Model 4-IA		−0.0130	−0.166	−0.096	0.002	−0.134	−0.173	−0.105	0.001
Model 5-IA		−0.055	−0.082	−0.033	0.001	−0.059	−0.084	−0.039	0.001
Model 6-IA		−0.107	−0.140	−0.077	0.002	−0.108	−0.140	−0.080	0.001
Model 7-SMA and IA	SMA	−0.121	−0.149	−0.091	0.002	−0.132	−0.162	−0.104	0.002
	IA	−0.090	−0.119	−0.063	0.002	−0.092	−0.121	−0.068	0.001

Note: CI = confidence interval, LBC = lower bound, UBC = upper bound; Model 1-SMA: Social media addiction → Social Capital and mindfulness → Mental health (see Figure 2 and Figure 3). Model 2-SMA: Social media addiction → Social Capital → Mental health (see Section A.1 and Section B.1). Model 3-SMA: Social media addiction → mindfulness → Mental health (see Section A.2 and Section B.2). Model 4-IA: Internet addiction → Social Capital and mindfulness → Mental health (see Figure 4 and Figure 5); Model 5-IA: Internet addiction → Social Capital → Mental health (see Section A.3 and Section B.3); Model 6-IA: Internet addiction → mindfulness → Mental health (see Section A.4 and Section B.4); Model 7-SMA and IA: Social media addiction and Internet addiction → Social Capital and mindfulness → Mental health (see Figure 6 and Figure 7).

## Data Availability

The raw data supporting the study of this article is publicly attached with this article.

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
