# Peer review of "A Multi- Mediation Analysis on the Impact of Social Media and Internet Addiction on University and High School Students’ Mental Health Through Social Capital and Mindfulness"

_ijerph, 2025, doi:10.3390/ijerph22010057_

Round 1

Reviewer 1 Report

Comments and Suggestions for Authors

I strongly believe that the relationship among internet addiction , social media addiction , social capital, mindfulness, and mental health is an important topic to examine. Also I consider it important to carry out this research in the age we live in the information age and experience intense technology-social media addiction.  This study aims to examine  the connections between IA, SMA, social capital, mindfulness, and mental health and adopts a serial mediation analysis.

The paper presents a detailed literature framework and explains all the variables in the research.  It must be acknowledged that the literature review is somewhat more extensive than is ideal, but it is, on the whole, satisfactory. 

It is beneficial to have a comprehensive understanding of the measurement tools utilized in the research. The methodology employed is clearly delineated and documented. I was nice to see Confirmatory Factor Analiysis (CFA) and AVE analysis in the methodology section.

The authors should clarify a few issues in the paper. First, they should provide more information about the sampling method. I could not find any details about how the sample was chosen. It would be helpful to know whether the authors used convenient sampling or another method.

Second, the authors should explain why they included high school and university students in the research. These two groups of students are believed to have different motivations and aims when using social media.  

Thirdly, based on the findings of this research, what action should be taken? What recommendations would you propose to educators, government officials and policy makers? Please clarify theese in your paper.

Fourthly, the length of some tables in the study was somewhat excessive, which was due to the nature of the research. I am aware that there is little that the authors can do about this and that I am not in a position to offer any suggestions. However, I felt it was important to draw this to their attention. 

In some parts of the study, the fonts were different. This should be checked.

The summary part of the study needs to be reconsidered and rewritten. There are too many numerical values in the summary. It should be simplified by expressing it in a more understandable and verbal format. Apart from these issues, I believe the study is valuable and will contribute to the field.

Author Response

Response to Reviewer's Comments (Reviewer 1)

Comment: I strongly believe that the relationship among internet addiction, social media addiction, social capital, mindfulness, and mental health is an important topic to examine. Also, I consider it important to carry out this research in the age we live in the information age and experience intense technology-social media addiction.  This study aims to examine the connections between IA, SMA, social capital, mindfulness, and mental health and adopts a serial mediation analysis. The paper presents a detailed literature framework and explains all the variables in the research.  It must be acknowledged that the literature review is somewhat more extensive than is ideal, but it is, on the whole, satisfactory. It is beneficial to have a comprehensive understanding of the measurement tools utilized in the research. The methodology employed is clearly delineated and documented. I was nice to see Confirmatory Factor Analysis (CFA) and AVE analysis in the methodology section.

Response: Dear reviewer, our study aims to shed light on the intricate interplay between IA, SMA, social capital, mindfulness, and mental health through a multi- mediation analysis. We are pleased that you found our literature framework detailed and comprehensive, providing a thorough explanation of the variables under investigation. Acknowledging your observation regarding the extensive nature of the literature review, we have sought to strike a balance between depth and conciseness in subsequent revisions while ensuring that the essential aspects are retained for a robust theoretical foundation. That means we have improved and reduce some redundant and unnecessary information and incorporated our refined issues in our manuscript highlighted in blue color.

We also agree on the significance of understanding the measurement tools employed, and we have endeavored to enhance clarity in this regard. We are encouraged by your positive remarks on the clarity and documentation of the methodology, particularly the inclusion of Confirmatory Factor Analysis (CFA) and Average Variance Extracted (AVE) analysis, which contribute to the rigor construct validity of our research findings. Your feedback has been invaluable in improving our work, and I hope we may address your concerns int this revised version attched.

Comment: The authors should clarify a few issues in the paper. First, they should provide more information about the sampling method. I could not find any details about how the sample was chosen. It would be helpful to know whether the authors used convenient sampling or another method.

Response: We acknowledged the need for more transparency regarding our sampling method. In our revised manuscript, we have included a detailed description of the sampling process. Specifically, the university and high schools were purposefully selected due to their convenience for data collection and their large populations. The samples were then chosen using proportionate stratified random sampling. For instance, the high school student samples were selected across grades and genders (refer to Table 1) based on the population numbers. After clearly defining the strata, we utilized simple random sampling to select the samples (kindly refer page 13 of the revised manuscript).

Comment: Second, the authors should explain why they included high school and university students in the research. These two groups of students are believed to have different motivations and aims when using social media.  

Response: Dear reviewer, we really appreciated your insightful comments. We recognized the importance of explaining our decision to include both high school and university students. In the revised manuscript, we have elaborated on this point, noting that these groups often exhibit distinct patterns of social media usage and motivations. Understanding these differences allows for a more nuanced analysis of the impact of social media on mental health across different educational stages.

Comment: Thirdly, based on the findings of this research, what action should be taken? What recommendations would you propose to educators, government officials and policy makers? Please clarify these in your paper.

Response: Dear reviewer we relay appreciated your scientific comments. We believed that your comment helps us to improve the quality of our paper and following your suggestions the following recommendations have included in the manuscript under discussion section.

Overall, the research on the connections among internet addiction (IA), social media addiction (SMA), social capital, mindfulness, and mental health addresses a pressing issue in today’s digital landscape. Based on the major findings, we suggest the following recommendations for educators, researchers and policymakers:               

  Recommendations for Researchers

  • Further Research on Intervention Efficacy: Longitudinal studies should be conducted to establish the long-term efficiency of both mindfulness and social capital interventions in reducing SMA and IA and further improving mental health outcomes for all age groups.
  • Investigation across Cultures: Understand how cultural variants influence the associations between SMA, IA, mindfulness, social capital, and mental health to aid in developing culturally sensitive interventions.
  • Specific Population Study: Focus on cohorts within university and high school settings to better understand the interaction between age, sex, and academic achievement and SMA, IA, mindfulness, and social capital.

Recommendations for Educators

  • Engage Students in Mindfulness Practices: We recommended the Include mindfulness practices in the curriculum that will help students develop attention control, emotional regulation, and stress management skills.
  • Facilitate Positive Behavior Online Guide: Students in responsible behavior online, on good digital etiquette, and maintaining a healthy balance between activities online and offline.

Recommendations for Policy Makers

  • Digital Wellness Policies: Make policies that foster digital wellness; include guidelines with schools on actions to take on the challenges of SMA and IA among students.
  • Training Programs: Design training programs and allocate resources that would lead educators to know the ways of addressing issues related to digital addiction and mental health in educational settings. By pursuing such undertakings and recommendations, stakeholders will be able to consider a supportive environment, foster mental well-being, and build resiliency in facing challenges brought about by social media and internet addiction.
  • Digital Literacy Enhancement Facilitate the running of digital literacy programs that help students use social media and the internet appropriately and in a balanced manner in order to avoid addiction.
  • School-Based Interventions: Introduce the school-based intervention that focuses on developing resilience, coping strategies, and social skills, which are believed to offer resistance to negative effects related to SMA and IA.

Comment: Fourthly, the length of some tables in the study was somewhat excessive, which was due to the nature of the research. I am aware that there is little that the authors can do about this and that I am not in a position to offer any suggestions. However, I felt it was important to draw this to their attention. 

Response: We appreciate the feedback regarding the length of some tables. While the complexity of our data necessitated detailed presentations, we have accepted your concern and share fully. However, the complexities of the problem and the nature of study (the study groups: high school and university) necessitated detailed presentations. We hope you may understand our concern.

Comment: In some parts of the study, the fonts were different. This should be checked. The summary part of the study needs to be reconsidered and rewritten. There are too many numerical values in the summary. It should be simplified by expressing it in a more understandable and verbal format. Apart from these issues, I believe the study is valuable and will contribute to the field.

Response: Dear reviewer, thank you for bringing the font inconsistency to our attention. We conducted a thorough review of the manuscript to ensure uniformity in font style and size throughout the document. We have also re-written the conclusion section. 

Reviewer 2 Report

Comments and Suggestions for Authors

Review Report

Title: A Serial Mediation Analysis on the Impact of Social Media and Internet Addiction on University and High School Students’ Mental Health through the Mediating effects of Social Capital and Mindfulness

Manuscript ID: ijerph-3133496

This research explores the impact of internet addiction (IA) and social media addiction (SMA) on the mental health of Ethiopian youth, with mindfulness and social capital serving as mediators. While the study is relevant and timely, several critical issues need to be addressed:

  1. The authors apply Positive Psychology and Social Cognitive Theory (SCT) to explain their findings. However, there seems to be a disconnect between these frameworks and the concepts central to the study. Seligman’s Positive Psychology Theory focuses on well-being through the PERMA model, which includes positive relationships but not mindfulness or social capital as core components. Similarly, Bandura’s Social Cognitive Theory emphasizes observational learning and self-efficacy, but does not explicitly highlight social capital or mindfulness as mechanisms for reducing IA or SMA. The manuscript lacks a clear and cohesive explanation of how these theories align with the proposed mediating roles of mindfulness and social capital. More clarity is needed on how the authors have theoretically connected these constructs to their study’s framework.
  2. The literature review highlights the effects of IA and SMA on mental health but fails to provide a detailed explanation of the mechanisms behind these effects as suggested by previous research. It would have been more valuable if the authors had synthesized existing literature to explain why and how IA and SMA negatively impact mental health and how mindfulness and social capital could mediate these relationships. The review also appears redundant in places, making it overly long and difficult to follow. A more concise and focused review that emphasizes the mechanisms and mediators would improve readability and understanding.
  3. In Line 361, the authors mention an "innovative approach," but it remains unclear what this innovation entails. The manuscript would benefit from a clearer explanation of the specific innovation the authors are referring to, especially in the context of their methodology.
  4. The authors claim to use a serial mediation model (even the research title suggests so), but the statistical analysis and the figure provided do not support this. Serial mediation implies a specific direction and flow between mediators (i.e., one mediator influencing the next), which is not evident in the provided analysis. Instead, the model appears to involve parallel mediation, where mindfulness and social capital act independently to mediate the relationship between IA, SMA, and mental health. The authors should reconsider their analysis and terminology to accurately reflect the statistical approach used.
  5. The study lacks clear hypotheses, which are essential for guiding the research and understanding the rationale behind the seven models tested. Without hypotheses, it is difficult to assess why so many models were created or whether the authors intended to compare the models. The development of specific, testable hypotheses based on the literature and the research questions would significantly improve the manuscript’s structure and coherence.
  6. The authors should also provide the age differences between high school and university students. It should be included as a factor in the demographics table and the correlation table.
  7. The manuscript does not address how the authors handled missing data, which is crucial for ensuring the validity and reliability of the findings. Clarification on the methods used to manage any missing data is necessary.
  8. As the study uses a cross-sectional design, the authors should explicitly mention this as a limitation. While mediation analysis can suggest indirect relationships, the inability to infer causality is a significant limitation.

Minor Comments:

  1. Several places contain typographical errors (e.g., "internet addtion" or "social media addition" instead of "internet addiction" or “social media addiction”). These should be corrected.
  2. The excessive length of the literature review, combined with some redundant information, makes it difficult to follow. A more concise and targeted approach would improve the paper.

Conclusion:

Overall, the study offers important insights into the mental health impacts of internet and social media addiction in a non-WEIRD cultural context, with a large sample size adding to its robustness. However, theoretical clarity, research hypothesis development, and methodological accuracy (particularly regarding the mediation approach) need significant revisions. Clarifying the statistical model and providing stronger theoretical justifications will strengthen the manuscript’s contribution to the literature on digital addiction and mental health.

In addressing these concerns, the study's overall clarity and rigor would be significantly improved, contributing to a more robust and comprehensible research endeavor. Without these clarifications, the paper is not accepted in its current state.

Author Response

Authors Response to the Reviewer 2

Comment 1: This research explores the impact of internet addiction (IA) and social media addiction (SMA) on the mental health of Ethiopian youth, with mindfulness and social capital serving as mediators. While the study is relevant and timely, several critical issues need to be addressed: The authors apply Positive Psychology and Social Cognitive Theory (SCT) to explain their findings. However, there seems to be a disconnect between these frameworks and the concepts central to the study. Seligman’s Positive Psychology Theory focuses on well-being through the PERMA model, which includes positive relationships but not mindfulness or social capital as core components. Similarly, Bandura’s Social Cognitive Theory emphasizes observational learning and self-efficacy but does not explicitly highlight social capital or mindfulness as mechanisms for reducing IA or SMA. The manuscript lacks a clear and cohesive explanation of how these theories align with the proposed mediating roles of mindfulness and social capital. More clarity is needed on how the authors have theoretically connected these constructs to their study’s framework.

Response: Theoretical Framework Connection: We appreciate your observation regarding the alignment of Positive Psychology and Social Cognitive Theory (SCT) with our study's constructs. In the revised manuscript, we have clarified how these theories and other relevant theories relate to our proposed mediators, mindfulness and social capital. In the revised manuscript we have incorporated the following.

The positive psychology theory proposed by Seligman [139], the social cognitive theory developed by [140], transactional model of stress and coping (TMSC) by [141]; and the cognitive-behavioral model of pathological internet use [142-143] serve as the theoretical foundations for this novel study. The theory of positive psychology, as developed by [139], has been associated with various phenomena such as Internet addiction, social media addiction, mindfulness, social capital, and mental health. Consequently, positive psychology theory offers a practical framework for examining the role of mindfulness as mediator in the relationship among IA, SMA, and the mental health of students. Nowadays, due to its highly beneficial qualities mindfulness construct had integrated with the positive psychology model [144] and mindfulness based programs and practices have been highly associated with many positive mental health and psychological outcomes [145]. Additionally, Albert Bandura's Social Cognitive Theory [140] explains that regulating social activities, including healthy social media use and internet access, plays a crucial role in regulating social and moral behavior as well as essential resources for social functioning, fostering the development of social competencies and internal standards. Inadditon, the transactional model of stress and coping (TMSC) by [141] posits that individuals' response to stress involves transactions between the person and their environment and it could be relevant for understanding how mindfulness and social capital act as coping mechanisms in dealing with stressors related to IA and SMA [141]. The theory also provides the best framework for investigating how individuals' coping strategies, including mindfulness practices and social capital (social support networks), influence their experiences with addictive behaviors [141].

Finally, a cognitive-behavioral model of pathological internet use (PIU) by [142] theorizes that interent addiction links cognitive symptoms such as ruminative cognitive styles, low self-worth, and social anxiety to the development of internet addiction, particularly in the context of social media use [142]. This model suggests that these cognitive factors can contribute to excessive and problematic online behaviors, affecting social capital by potentially reducing face-to-face interactions and impacting mental health through heightened feelings of loneliness, isolation, and diminished self-esteem, emphasizing the need for interventions that address these cognitive patterns to promote healthier internet use and improve overall well-being internet [6, 142-143]. The theoretical models used in this study such as the Seligman's positive psychology theory [139], Bandura's social cognitive theory [140], the transactional model of stress and coping (TMSC) [141], and a cognitive-behavioral model of Pathological Internet Use (PIU [142] emphasizes that social networking, social capital and mindfulness are crucial components for enhancing mental health and reducing internet and social media addiction among university and high school students [140-143]. These theories emphasize the importance of person-environment interaction, social net-working, being aware of the environment, social capital and mindfulness as positive psychological and social resources for reducing internet and social media addiction and promoting mental health. Regarding the positive psychology construct yes, PERMA Profiler is the most recognized model. However, nowadays due to their benefits and high relationships different constructs incorporated with positive psychology. We emphasize that while Seligman’s PERMA model focuses on well-being, mindfulness is a key practice that enhances emotional regulation and resilience, which aligns with the well-being aspect of Positive Psychology. Similarly, we discuss how social capital can be understood through the lens of SCT, as it relates to social learning and community support, which are crucial for mitigating the impacts of IA and SMA. We have mentioned above some relevant references how mindfulness connected with positive psychology.   

Comment 2: The literature review highlights the effects of IA and SMA on mental health but fails to provide a detailed explanation of the mechanisms behind these effects as suggested by previous research. It would have been more valuable if the authors had synthesized existing literature to explain why and how IA and SMA negatively impact mental health and how mindfulness and social capital could mediate these relationships. The review also appears redundant in places, making it overly long and difficult to follow. A more concise and focused review that emphasizes the mechanisms and mediators would improve readability and understanding.

Response: Mechanisms of IA and SMA on Mental Health: We recognize the need for a more detailed synthesis of the mechanisms through which IA and SMA affect mental health. The literature review has been revised to include comprehensive explanations of these mechanisms, supported by recent studies. We have focused on how these addictions can lead to social isolation and anxiety, and how mindfulness and social capital may serve as protective factors Kindly refer the revised version highlighted in blue color).

Comment 3: In Line 361, the authors mention an "innovative approach," but it remains unclear what this innovation entails. The manuscript would benefit from a clearer explanation of the specific innovation the authors are referring to, especially in the context of their methodology.

Response: Dear reviewer, we thank you for your constructive comment. In this study, we have provided a clearer explanation of what we mean by "innovative approach." This includes specifying our methodological, conceptual and contextual contributions, such as the use of a combined multi- mediation analysis with a focus on both mindfulness and social capital as mediators, which is relatively underexplored in existing literature. Also, this study tried to show the new conceptual study have never studied before in this way and in Ethiopian context no single study focused on these issues in comprehensive way.

Comment: The authors claim to use a serial mediation model (even the research title suggests so), but the statistical analysis and the figure provided do not support this. Serial mediation implies a specific direction and flow between mediators (i.e., one mediator influencing the next), which is not evident in the provided analysis. Instead, the model appears to involve parallel mediation, where mindfulness and social capital act independently to mediate the relationship between IA, SMA, and mental health. The authors should reconsider their analysis and terminology to accurately reflect the statistical approach used.

Response: Dear reviewer we really appreciated for your valuable comment for highlighting the inconsistency in our mediation model. We thought Internet addiction and SMA covariate considered as one path. However, that was correlation among the constructs.   We have revised the title that fit with our statistical analysis and terminology to accurately reflect our approach “A Multi- Mediation Analysis on the Impact of social media and Internet Addiction on University and High School Students’ Mental Health through Social Capital and Mindfulness: We hope your question may be answered.  

Comment: The study lacks clear hypotheses, which are essential for guiding the research and understanding the rationale behind the seven models tested. Without hypotheses, it is difficult to assess why so many models were created or whether the authors intended to compare the models. The development of specific, testable hypotheses based on the literature and the research questions would significantly improve the manuscript’s structure and coherence.

Response:  Dear reviewer we appreciated your inquiry regarding the development of hypotheses instead of research questions. In response to your comment about the lack of clear hypotheses, due to the complexity nature of the study we have large number of hypotheses.  As a result, we designed only few research question that in compass different hypotheses that guide our research and clarify the rationale behind our models. This may be strengthening the manuscript’s structure and coherence, making it easier for readers to understand the research objectives. But we highly appreciated and acknowledge your comment.

Comment: The authors should also provide the age differences between high school and university students. It should be included as a factor in the demographics table and the correlation table.

Response: Dear reviewer We have included the age differences between high school and university students in the demographics table and the correlation tableThe demographic questionnaire obtained information regarding sex, age, grade level, and batch (year) of the students. Also, the correlation of these factors included under “Table 3. Pearson correlations (r) among the socio-demographic factors and the study variables”. After checking their relationship, we have continued further analysis. We hope your query may be answered.

Comment: The manuscript does not address how the authors handled missing data, which is crucial for ensuring the validity and reliability of the findings. Clarification on the methods used to manage any missing data is necessary.

Response: Dear reviewer We appreciate your reminder regarding the handling of missing data. A=Initially response dents we fill inappropriate data discarded not considered at all. To increase the accuracy of the data there were four attention and honesty check items included based on scientific recommendations [147-148]. The first item, "Please choose 'agree' for this question," was used to test whether the subject was careless or inattentive. If the subject did not choose the fixed response of "agree," their data was excluded [147-149]. The next three items which were rated on a 4-point Likert scale from 1 (strongly disagree) to 4 (strongly agree)- "I answered all the questions truthfully," "I never lied," and "I never hid myself" - were intended to assess the honesty and truthfulness of the subject's responses. If a person scored low (≤2 on the 4-point Likert scale) on any of these three items, their data was also excluded, as this would suggest a tendency toward self-reported deception [147-149]. By including these validity checks, the researchers aimed to ensure the truthfulness and reliability of the self-reported data collected through the demographic questionnaire.

Comment: As the study uses a cross-sectional design, the authors should explicitly mention this as a limitation. While mediation analysis can suggest indirect relationships, the inability to infer causality is a significant limitation.

Response: Dear reviewer, we have explicitly stated the limitations of our cross-sectional design in the revised manuscript. We acknowledge that while mediation analysis can suggest indirect relationships, the inability to infer causality is a significant limitation that readers should be aware of. Hence, we have included it in the limitation section. Here is the limitation included in the manuscript: Finally, while this study utilizes mediation analysis to explore potential indirect relationships among variables, it is important to acknowledge a significant limitation due to the inability to infer causality. Mediation analysis can indicate associations and suggest pathways, but it does not establish definitive cause-and-effect relationships. Therefore, we recommend that future studies conduct experimental and quasi-experimental designs.

Minor Comments:

Comment: Several places contain typographical errors (e.g., "internet addtion" or "social media addition" instead of "internet addiction" or “social media addiction”). These should be corrected.

Response: Dear reviewer we have revised and corrected all typographical errors throughout the manuscript.

Comment: The excessive length of the literature review, combined with some redundant information, makes it difficult to follow. A more concise and targeted approach would improve the paper.

Response: Dear reviewer we have removed the redundant sections of the literature review and carefully revised.

Comment:  Conclusion: Overall, the study offers important insights into the mental health impacts of internet and social media addiction in a non-WEIRD cultural context, with a large sample size adding to its robustness. However, theoretical clarity, research hypothesis development, and methodological accuracy (particularly regarding the mediation approach) need significant revisions. Clarifying the statistical model and providing stronger theoretical justifications will strengthen the manuscript’s contribution to the literature on digital addiction and mental health. In addressing these concerns, the study's overall clarity and rigor would be significantly improved, contrbuting to a more robust and comprehensible research endeavor. Without these clarifications, the paper is not accepted in its current state.

Response: Dear reviewer, we appreciate the constructive comments and are dedicated to making the necessary improvements to ensure that the study meets the required quality standards for publication. Thank you for the valuable feedback, and we look forward to resubmitting an improved version of the manuscript for further consideration. In conclusion, we have made substantial revisions in response to your valuable feedback. We believe that these changes have significantly improved the clarity, rigor, and theoretical grounding of our manuscript. Thank you once again for your insight and guidance. We hope that the revised manuscript meets your expectations and contributes meaningfully to the literature on digital addiction and mental health.

Reviewer 3 Report

Comments and Suggestions for Authors

The topic addressed by the authors entitled: “A Serial Mediation Analysis on the Impact of Social Media and 2 Internet Addiction on University and High School Students' 3 Mental Health through the Mediating effects of Social Capital 4 and Mindfulness” refer to issues of great importance for contemporary educational research.

In terms of its strength:

- It highlights the fact that it analyzes a phenomenon that needs to be further explored today. The manuscript provides relevant information for the topic, making it a strong point. It should also be noted that some recent references have been used together with other more classical ones, which helps to see the evolution of terms and the evolution of the subject. 

- The theoretical framework is well developed and structured (this brings more sense and structure to the literature of the study).

- The methodological assumptions used in this project are unobjectionable. The results of the study are clearly presented. There is also no doubt about the interpretation of the results.

- On the other hand, a “Discussion” and “conclusion” section is specified, giving greater rigor to the study.

- A section on limitations and future lines of research is applied, which gives the study greater scientific quality.

In the process of reviewing the study, some aspects have been identified that, if addressed, could strengthen the rigor and clarity of the study. The recommendations are detailed below:

- Research objectives: It is important to clearly identify the specific objectives of the research in relation to the subject matter addressed. This will help to better delimit the scope and purpose of the study.

- Formulation of hypotheses: Currently, no hypotheses have been published in the study. It is suggested to include them, since they are necessary for this type of research and could provide structure to the development of the analysis. 

- The bibliographic references applied are correct, adjusting to the current regulations, however, I recommend the authors to review some of them, since they have DOI and it is not reflected as other data.

Importantly, I recommend the authors to omit in the article those tables and figures that are not essential, since their inclusion may make the content excessively long, and may even create disinterest in the reader.

Finally, I congratulate the authors on their study and encourage them to implement these adjustments that will bring scientific quality to their research.

Author Response

Authors Response to Reviewer's Comments (No.3)

Comment: The topic addressed by the authors entitled: “A Serial Mediation Analysis on the Impact of Social Media and 2 Internet Addiction on University and High School Students' 3 Mental Health through the Mediating effects of Social Capital and Mindfulness” refer to issues of great importance for contemporary educational research.

Response: We would like to express our gratitude for your positive feedback and constructive suggestions regarding our manuscript titled “A Serial Mediation Analysis on the Impact of Social Media and Internet Addiction on University and High School Students' Mental Health through the Mediating Effects of Social Capital and Mindfulness.” Your insights are invaluable in enhancing the quality and clarity of our research. Below, we address your recommendations:

Comment: In terms of its strength, It highlights the fact that it analyzes a phenomenon that needs to be further explored today. The manuscript provides relevant information for the topic, making it a strong point. It should also be noted that some recent references have been used together with other more classical ones, which helps to see the evolution of terms and the evolution of the subject. 

  • The theoretical framework is well developed and structured (this brings more sense and structure to the literature of the study).
  • The methodological assumptions used in this project are unobjectionable. The results of the study are clearly presented. There is also no doubt about the interpretation of the results.
  • On the other hand, a “Discussion” and “conclusion” section is specified, giving greater rigor to the study.
  • A section on limitations and future lines of research is applied, which gives the study greater scientific quality.
  • In the process of reviewing the study, some aspects have been identified that, if addressed, could strengthen the rigor and clarity of the study. The recommendations are detailed below: 

Response: Dear reviewer, Dear Reviewer,

Thank you for your positive feedback. We appreciate your acknowledgment of our work. The theoretical framework is well-developed and structured, providing clarity and coherence to the literature of the study. The methodological assumptions used in this project are sound, and the results are clearly presented. There is also no doubt regarding the interpretation of the results. Additionally, the inclusion of a "Discussion" and "Conclusion" section adds rigor to the study. A section on limitations and future lines of research has been included, which enhances the scientific quality of the study.

Research objectives: It is important to clearly identify the specific objectives of the research in relation to the subject matter addressed. This will help to better delimit the scope and purpose of the study.

Response: Dear reviewer, we appreciate your suggestion to clearly delineate the specific objectives of our research. In the revised manuscript, we have added a section explicitly outlining the research objectives, which helps to clarify the scope and purpose of our study. Below are our objectives incorporated in the revised manuscript.

Objective

General Objective

  • The main aim of this study was to examine the complex associations that exist between social media addiction, internet addiction, mindfulness, social capital, and mental health for university and high school students by investigating the mediating influence of social capital and mindfulness.

Specific Objectives

  • To test the relationship that may exist between social media addiction, internet addiction, social capital, mindfulness, and mental health for university and high school students.
  • To assess the direct effects of social media addiction and Internet addiction on social capital, mindfulness, and mental health. To explore the positive and direct impacts of social capital and mindfulness on the mental health of students. To see whether social capital and mindfulness partly mediate the association between social media addiction and mental health and between Internet addiction and mental health.
  • To assess the degree to which social capital and mindfulness are complete or partial mediators in the relationships of social media addiction, internet addiction, and mental health constructs.
  • To suggest possible recommendations and inform policymakers, educators, and mental health professionals about better interventions and strategies to fostering healthy digital habits, thereby enhancing mental well-being among students in a digital era.

Comment:   Formulation of hypotheses: Currently, no hypotheses have been published in the study. It is suggested to include them, since they are necessary for this type of research and could provide structure to the development of the analysis. 

Response: Dear reviewer, we acknowledge the importance of including hypotheses in our research. However, instead of numerous research hypotheses, we adopted a few comprehensive research questions that address our research objectives. We appreciate your understanding of our concerns regarding the complex nature of the study.

Comment:  The bibliographic references applied are correct, adjusting to the current regulations, however, I recommend the authors to review some of them, since they have DOI and it is not reflected as other data.

Response: Dear reviewer we thank you and appreciated for your constructive comments. Regarding your valuable suggestions, we have conducted a thorough review and ensured that all references are consistent with current citation standards, including the addition of DOIs where applicable.

Comment:  Importantly, I recommend the authors to omit in the article those tables and figures that are not essential, since their inclusion may make the content excessively long, and may even create disinterest in the reader.

Response: Dear Reviewer, due to the complex nature of our study, we have made efforts to enhance its appeal and clarity in the tables and figures. Initially, we removed 6 tables and 5 figures, and we have attached some of them in the supplementary files

Comment:  Finally, I congratulate the authors on their study and encourage them to implement these adjustments that will bring scientific quality to their research.

Response: Dear reviewer, we are grateful for your acknowledgment of the strengths of our theoretical framework, methodology, and the inclusion of discussions on limitations and future research. We have made minor improvements in these sections based on your feedback to further enhance the rigor of the study.

In conclusion, we are thankful for your encouragement and constructive suggestions. We believe that the adjustments made in response to your comments will significantly improve the clarity and scientific quality of our research. We look forward to your further feedback and hope that the revised manuscript meets your expectations

Once again, we appreciate your valuable feedback, and we believe that these revisions have significantly enhanced the clarity and effectiveness of our article. We look forward to your feedback on the updated version.

Reviewer 4 Report

Comments and Suggestions for Authors

Thank you for the opportunity to review this manuscript on such a topical and important subject.

The manuscript describes all the parts of the study very well, but I think the introduction is too long and becomes repetitive. The description of the collection instruments is also too long; I don't think such an exhaustive description and explanation is necessary.

In the introduction, when describing some studies, you should refer to the author(s) and the study carried out and not just the number of the bibliographic reference (Ex lines 138, 139 and 140).

In general, I think the whole manuscript should be revised in order to better summarize all the chapters.

I have no further suggestions.

Author Response

Authors Response for Reviewer 4

Comment: Thank you for the opportunity to review this manuscript on such a topical and important subject.

Response: We sincerely appreciate your thoughtful review and valuable feedback on our manuscript. Your insights have helped us identify areas for improvement, and we are grateful for the opportunity to enhance our work. Below, we address your specific comments:

Comment: The manuscript describes all the parts of the study very well, but I think the introduction is too long and becomes repetitive. The description of the collection instruments is also too long; I don't think such an exhaustive description and explanation is necessary.

Response: Dear reviewer, we acknowledge and appreciated your concern regarding the length and repetitiveness of the introduction. In the revised manuscript, we have streamlined the introduction by removing redundant information and focusing on key points that directly support the study's objectives. This should enhance clarity and maintain reader engagement. Attached below our revision.

Comment: In the introduction, when describing some studies, you should refer to the author(s) and the study carried out and not just the number of the bibliographic reference (Ex lines 138, 139 and 140).

Response: Thank you for your suggestion to refer to the author(s) and their studies rather than just bibliographic references. We appreciated your valuable comment, however we strictly followed the journal guideline. We appreciate again for your suggestions.

Comment: In general, I think the whole manuscript should be revised in order to better summarize all the chapters. I have no further suggestions.

Response: Dear reviewer, we have conducted a thorough review of the entire manuscript to identify areas where summarization is needed. We have worked to condense sections across the manuscript to ensure that each chapter is clearly and succinctly presented.

Once again, we appreciate your valuable feedback, and we believe that these revisions have significantly enhanced the clarity and effectiveness of our article. We look forward to your feedback on the updated version.

Round 2

Reviewer 2 Report

Comments and Suggestions for Authors

I have reviewed the authors' responses to my queries. While they have made an effort to address the concerns raised, I still believe the paper's most significant shortcoming lies in the absence of clear hypotheses. For a study employing a complex multi-mediation model with a quantitative approach, it is essential to establish clear hypotheses. If the authors' intent is exploratory, this should be made explicit

Additionally, the authors have introduced a new objective to provide recommendations and inform policymakers. However, this objective is not supported by any policy framework, nor are there relevant references to justify this aim. While recommendations have been included in the revised version, they appear misaligned with the study's focus. For example, the suggestion for school-based interventions seems out of place, as the study does not address a school-based context. Similarly, the recommendation to create programs for healthy digital well-being to educate parents seems disconnected from the research, as parental factors are not included in the study.

Overall, the recommendations at the end appear random and lack coherence, especially within the context of scientific research. In its current form, I would suggest the authors to reconsider the inclusion of the policy framework objective and significantly revise or eliminate the recommendations (perhaps condensing them to one or two relevant points).

Author Response

Dear reviewer, in response to your review, we have thoroughly considered your comments and revised the paper according to your valuable recommendations. First, We have successfully integrated the research hypotheses into each subsection of the literature, ensuring a coherent structure throughout. Your feedback has been incredibly insightful and constructive.

Second, concerning the recommendations, we have taken great care to carefully outline them based on recent, reputable sources that align closely with the research aim. The previous recommendations have been entirely revised and are now supported by empirical evidence.

We hope that these revisions effectively address your concerns. We sincerely appreciate your tireless efforts in providing such helpful scientific suggestions. Our responses have been highlighted in red throughout the manuscript. Thank you for your invaluable comments.
